# A Review of Parallel Robots: Rehabilitation, Assistance, and Humanoid Applications for Neck, Shoulder, Wrist, Hip, and Ankle Joints

**Victoria E. Abarca** *,† and **Dante A. Elias** †

Biomechanics and Applied Robotics Research Laboratory, Pontificia Universidad Católica del Perú, Lima 15088, Peru; delias@pucp.pe
* Correspondence: victoria.abarca@pucp.edu.pe
† These authors contributed equally to this work.

**Abstract:** This review article presents an in-depth examination of research and development in the fields of rehabilitation, assistive technologies, and humanoid robots. It focuses on parallel robots designed for human body joints with three degrees of freedom, specifically the neck, shoulder, wrist, hip, and ankle. A systematic search was conducted across multiple databases, including Scopus, Web of Science, PubMed, IEEE Xplore, ScienceDirect, the Directory of Open Access Journals, and the ASME Journal. This systematic review offers an updated overview of advancements in the field from 2012 to 2023. After applying exclusion criteria, 93 papers were selected for in-depth review. This cohort included 13 articles focusing on the neck joint, 19 on the shoulder joint, 22 on the wrist joint, 9 on the hip joint, and 30 on the ankle joint. The article discusses the timeline and advancements of parallel robots, covering technology readiness levels (TRLs), design, the number of degrees of freedom, kinematics structure, workspace assessment, functional capabilities, performance evaluation methods, and material selection for the development of parallel robotics. It also examines critical technological challenges and future prospects in rehabilitation, assistance, and humanoid robots.

**Keywords:** assistance; exoskeletons; parallel robots; prosthetics; rehabilitation

## 1. Introduction

Rehabilitation and assistance for human body joints play crucial roles in people's health, well-being, and quality of life. These activities are essential for helping individuals recover or improve their functionality, mobility, and autonomy following injuries, surgeries, or diseases affecting the joints. In recent years, technological advances have opened up new possibilities in the use of parallel robots in the fields of rehabilitation, assistive technologies, and humanoid systems. These robotic systems are specifically designed to facilitate the recovery of motor and functional skills. Featuring a parallel mechanical structure, these robots offer greater precision, stability, and adaptability to meet the individual rehabilitation or assistance needs of patients.

Rehabilitation technologies aid in the recovery or improvement of motor function after an injury or illness. Utilizing parallel robots in rehabilitation enables specialists to help patients regain strength and mobility in affected joints more rapidly and efficiently than traditional therapeutic methods. The scope of these technologies includes rehabilitation devices for patients with head and neck injuries [1–4]; pediatric rehabilitation devices for the arm [5]; wearable rehabilitation devices for the arm [6]; exoskeletons for the arms of patients with stroke and spinal cord injuries [7]; and devices specifically designed for wrist [8–10], ankle [11–16], and foot rehabilitation [17].

Parallel robots in assistive devices help individuals carry out activities of daily life, thereby providing greater autonomy and independence. Examples include prostheses designed to offer functional mobility by replacing missing limbs in amputees [18], as well

as shoulder disarticulation arm prostheses [19,20], wrist prostheses [21], and disarticulated hip prostheses [22].

The integration of humanoid robots into medical rehabilitation and assistance offers exciting opportunities. These include personalized therapy, precise motivation and tracking, assisted mobility, and the objective assessment of patients' progress.

Parallel robots have advanced significantly in the fields of architectural [23] and mathematical modeling, particularly in kinematic [24–26] and dynamic analyses [27]. These robots can move at high speeds due to their lightweight and simple structural design [28]. They are also highly rigid, making them ideal for tasks requiring substantial force or pressure [29]. Their high-precision design minimizes unwanted movements and vibrations [30]. Furthermore, they can support heavier loads [31,32] and offer better positioning accuracy due to their high rigidity and low weight, which ensure minimal deformation [33]. Utilizing sensor technology and control algorithms, these parallel robots can automatically adjust to meet each patient's specific needs, whether for rehabilitation or assistance, thus delivering a highly personalized experience.

However, there is a gap in the current literature concerning the use of parallel robots in the fields of rehabilitation, assistance, and humanoid systems. This review aims to address this gap by summarizing the available evidence on the utilization of parallel robots.

Therefore, this article aims to review the state of parallel robot technology as applied to rehabilitation, assistance, and humanoid systems, focusing on joints with three degrees of freedom in the human body: the neck, shoulder, wrist, hip, and ankle. Initially, the search strategy, inclusion and exclusion criteria, quality assessment, and data extraction methods are defined. Subsequently, the search results are presented, along with a detailed description of the biomechanics of the joints, parallel robots, and applications in the medical field. The article then discusses the timeline and advancements of parallel robots between 2012 and 2023 and examines the technology readiness levels (TRLs), design, number of degrees of freedom, kinematics structure, workspace assessment, functional capabilities, performance methods, and material selection in the development of parallel robotics, as well as the critical technological challenges and future prospects in rehabilitation, assistance, and humanoids. Finally, conclusions are presented.

## 2. Methodology

### 2.1. Search Strategy

Before initiating the search, a research question was formulated: "Are there parallel robots designed for rehabilitation, assistance, and humanoid applications that target joints such as the neck, shoulder, wrist, hip, and ankle?" This question helped focus the search for pertinent information. Subsequently, several databases were identified for the search, including Scopus, Web of Science, PubMed, IEEE Xplore, ScienceDirect, the Directory of Open Access Journals (DOAJ), and the ASME Journal. Following this, search terms were determined, incorporating keywords related to the topic such as "parallel robot and neck", "parallel robot and shoulder", "parallel robot and hip", "parallel robot and wrist", and "parallel robot and ankle".

### 2.2. Inclusion Criteria

The search was conducted from 2012 to 2023, and articles were selected based on the following inclusion criteria:

- Articles in English that discuss parallel robots for the rehabilitation of the neck, shoulder, wrist, hip, and ankle.
- Articles in English that discuss parallel robots for assistance related to the neck, shoulder, wrist, hip, and ankle.
- Articles in English that discuss parallel robots for humanoid applications focusing on the neck, shoulder, wrist, hip, and ankle.
- Parallel robots with three or more degrees of freedom.

- Articles discussing parallel robots at either the conceptual or prototype level of technological development.

### 2.3. Exclusion Criteria

The article search, conducted between 2012 and 2023, was performed based on the following exclusion criteria:

- Articles that do not discuss parallel robots.
- Articles unrelated to rehabilitation involving parallel robots.
- Articles unrelated to assistance involving parallel robots.
- Articles unrelated to humanoid applications involving parallel robots.

### 2.4. Quality Assessment

Articles were identified and selected based on their relevance to the research question. Initially, the titles and abstracts were reviewed to determine if they met the inclusion criteria. Subsequently, the content of each article was thoroughly read. The quality of the selected articles was then evaluated based on the reputation and trustworthiness of the scientific journals in which they were published.

### 2.5. Data Extraction

Data from the selected articles were synthesized to answer the research question. This involved conducting an analysis of various factors concerning the development of parallel robots, including the year of publication, country of development, type of rehabilitation targeted at specific joints, level of technological maturity, types of mechanisms, degrees of freedom, types of movement, types of actuators, kinematic models, and simulation tools used.

### 2.6. Search Performance

The search yielded a total of 846 articles: Scopus contributed 128, Web of Science 103, PubMed 21, IEEE Xplore 63, ScienceDirect 271, DOAJ 101, and ASME Journal 159. After eliminating 23 duplicate articles, 823 articles remained. Following the application of the exclusion criteria, 93 papers were selected for review. These included 13 articles related to the neck joint, 19 related to the shoulder joint, 22 related to the wrist joint, 9 related to the hip joint, and 30 related to the ankle joint, as illustrated in Figure 1.

### 2.7. Systematic Review

The 93 selected articles were analyzed and categorized by joint type: neck, shoulder, wrist, hip, and ankle. Key aspects such as the year of publication, country of origin, type of rehabilitation targeted by joint, mechanism type, degrees of freedom, types of movement, type of actuator, mathematical models, and simulation tools were considered.

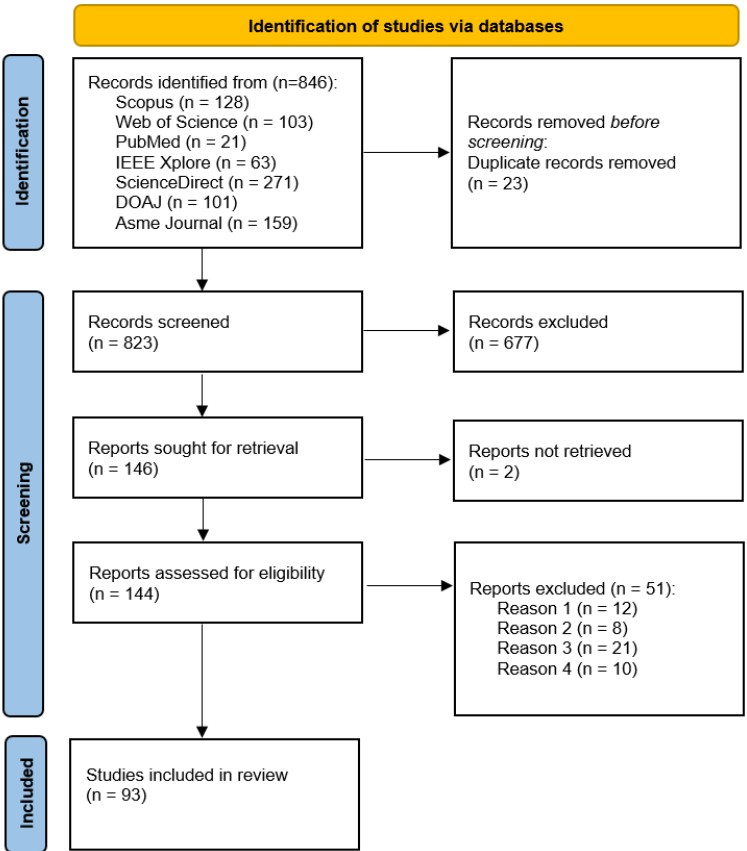

**Figure 1.** PRISMA flow diagram illustrating the application of inclusion and exclusion criteria to studies selected from 2012 to 2023, focusing on the shoulder, neck, wrist, hip, and ankle joints.

### 3. Biomechanics of the Human Joints

The relationship between the planes, axes, and ranges of motion in human joints is fundamental to understanding how joints function. These concepts are interrelated and play roles in describing and analyzing joint movement. Planes and axes of motion serve as useful tools for describing the directions and orientations of movements in the human body. The three primary planes, sagittal, frontal, and transverse, are associated with the frontal, sagittal, and vertical axes, respectively [34], as illustrated in Figure 2.

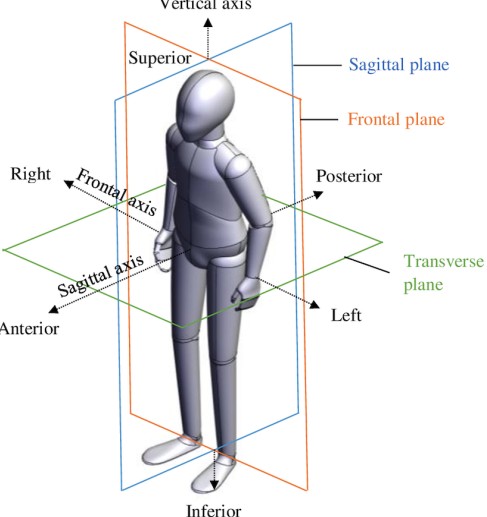

**Figure 2.** Anatomical sagittal, frontal, and transverse planes and vertical, sagittal, and frontal axes.

The range of motion is fundamental to the movement and functionality of the human body. The term "range of motion" refers to the various directions and amplitudes through which a joint or body segment can move. Biomechanics investigates not only how these movements occur but also how they are generated and influenced by factors such as anatomical structure, muscle strength, and external forces. This field of study also encompasses different types of joint movements, including flexion, extension, abduction, adduction, and rotation, describing these movements in three dimensions through a combination of planes and axes.

- Anatomical planes:
  Sagittal plane (or median plane): Divides the body into left and right halves. Movement in this plane is forward and backward.
  Frontal plane (or coronal plane): Divides the body into anterior (front) and posterior (back) halves. Movement in this plane is side-to-side.
  Transverse plane (or axial/horizontal plane): Divides the body into superior (upper) and inferior (lower) halves. Movement in this plane involves rotation.
- Anatomical axes:
  Sagittal axis (or anteroposterior axis): Extends front to back and is perpendicular to the frontal plane. Movements around this axis include abduction and adduction.
  Frontal axis (or horizontal axis): Extends side to side and is perpendicular to the sagittal plane. Movements around this axis include flexion and extension.
  Vertical axis (or longitudinal axis): Extends top to bottom and is perpendicular to the transverse plane. Movements around this axis include internal and external rotation.
- Range of motion of human joints:
  The range of motion (ROM) varies depending on the specific joint and is influenced by factors such as age, gender, flexibility, and the individual's physical condition. Table 1 presents the planes and axes of motion, types of motion, and ranges of motion for the neck, shoulder, hip, wrist, and ankle joints [34].

**Table 1.** Dominant movement, planes, axis, and range of motion (ROM) of the neck, shoulder, hip, wrist, and ankle joints.

| Joints | Movement | Plane | Axis | ROM (°) |
|---|---|---|---|---|
| Neck | Flexion/extension | Sagittal | Frontal | 0–35/0–45 |
| | Right/left bending | Frontal | Sagittal | 0–35/0–45 |
| | Right/left rotation | Transverse | Vertical | 0–60/0–80 |
| Shoulder | Flexion/extension | Sagittal | Frontal | 0–150/0–170 |
| | Abduction/adduction | Frontal | Sagittal | 0–160/0–30 |
| | Internal/external rotation | Transverse | Transverse | 0–70/0–70 |
| Hip | Flexion/extension | Sagittal | Frontal | 0–140/0–10 |
| | Abduction/adduction | Frontal | Sagittal | 0–50/0–30 |
| | Internal/external Rotation | Transverse | Vertical | 0–40/0–50 |
| Wrist | Flexion/extension | Sagittal | Frontal | 0–50/0–30 |
| | Abduction/adduction | Frontal | Sagittal | 0–25/0–30 |
| | Pronation/supination | Transverse | Vertical | 0–85/0–90 |
| Ankle | Plantarflexion/dorsiflexion | Sagittal | Frontal | 0–50/0–30 |
| | Abduction/adduction, | Frontal | Sagittal | 0–10/0–20 |
| | Inversion/eversion | Transverse | Vertical | 0–60/0–30 |

## 4. Parallel Robots and Applications in the Medical Field

Parallel robots have had a significant impact on the medical field, finding applications in areas such as rehabilitation, assistance, and humanoids.

- Rehabilitation robots are designed to assist individuals in regaining motor skills, functionality, and strength following injury or illness. Deployed in therapeutic settings,

these devices aim to enhance the individual's physical capabilities and long-term quality of life.

- Assistive robots are created to help individuals perform activities of daily living, such as dressing, running, or walking, when these activities are restricted due to disability or injury. These devices serve as aids to improve the individual's quality of life, offering autonomy and independence.

- Humanoid robots represent an impressive advancement in technology, designed to mimic and replicate human form and behavior. These robots are built with anatomical features resembling those of humans, including heads, torsos, arms, and legs, enabling them to move and execute tasks in a manner similar to humans.

Based on the preceding discussion, it is crucial to initially distinguish between parallel robots and their serial and hybrid counterparts. Following this differentiation, we can then delve into their applications in rehabilitation, assistance, and humanoid contexts, particularly focusing on specific joints like the neck, wrist, shoulder, hip, and ankle.

### 4.1. Distinctions Between Serial, Parallel, and Hybrid Robots

Serial robots have links and joints sequentially connected to maneuver the end-effector in relation to a stationary base. Parallel robots, on the other hand, feature several serial chains that link a movable platform to a fixed base using multiple independent kinematic chains. Hybrid robots integrate features from both, blending closed-chain structures with open-chain systems [21]. These distinctions relate directly to the robots' architectural design and intrinsic structure, as depicted in Figure 3.

Parallel robots vary in structure depending on the kinematic chain, which can consist of revolution (R), prismatic (P), universal (U), and spherical (S) type joints. The 3-SPS/S parallel mechanism serves as an example; the "3-SPS" portion of the name indicates the presence of three serial chains, each containing a spherical, prismatic, and then spherical joint. The "S" series chain, although consisting of only one spherical joint, remains parallel to the other three SPS chains [21].

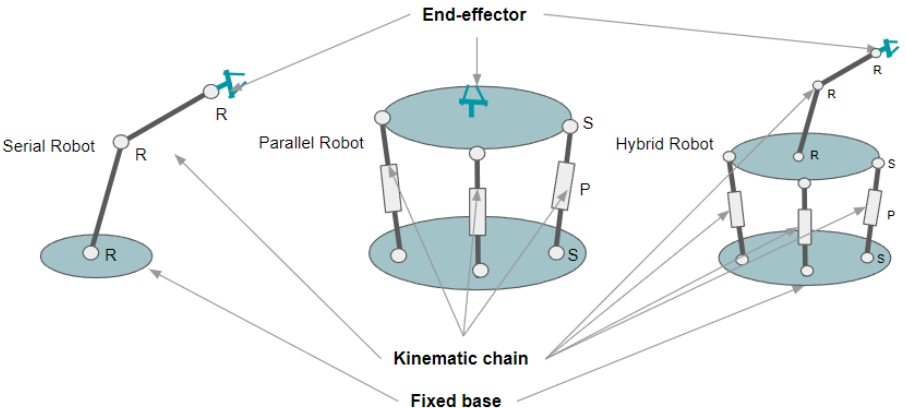

**Figure 3.** Architectural design of serial, parallel, and hybrid robots.

### 4.2. Parallel Robot for Neck Joint

Studies related to rehabilitation, assistance, and humanoid applications for the neck joint are summarized in Table 2.

#### 4.2.1. Parallel Robot for Neck Rehabilitation

In 2019, Limgampally et al. introduced a wearable therapy device for treating cervical spine injuries. The device used a three-degrees-of-freedom parallel manipulator and was intended for automated, safe operation under therapist supervision. Figure 4a shows the wearable therapeutic device designed to treat patients with head/neck posture disorders. It employs a 3-RPS parallel mechanism consisting of a movable platform (top) and a fixed platform, connected by three similarly designed supporting links. Each link sequentially



connects the top and fixed platforms through a revolute joint (R), a prismatic joint (P), and a spherical joint (S), with the prismatic joint being controlled by an electric linear actuator [1]. Zhang et al. also designed a dynamic neck brace used for characterizing head motion in ALS patients while simultaneously recording the surface electromyography (EMG) of neck muscles. The brace measured muscle activation and could supplement self-reported data to assess head drop and disease progression [2].

In 2022, Lozano et al. developed a closed-chain robotic active neck orthosis featuring four degrees of freedom, based on a four-legged Stewart platform configuration. The orthosis employed a robust control strategy with state restrictions, offering an innovative approach to treating neck ailments. The system was tested on selected volunteer subjects and successfully limited the range of motion within a pre-established area based on the patients' reported range of motion for conditions such as cervicalgia and whiplash syndrome [4].

In 2023, Zhang et al. presented a cable-driven exoskeleton specifically designed for cervical rehabilitation, addressing the urgent need given the increasing prevalence and younger onset of cervical conditions. The research began with an in-depth analysis of the mechanics of neck movement and the specific needs of rehabilitation, setting the foundation for the design criteria and the preliminary sketch of the exoskeleton. Subsequent phases involved kinematic modeling and simulation exercises, which not only confirmed the design's validity but also provided insights into cable adjustments across different rehabilitation paths [35].

### 4.2.2. Parallel Robot for Neck Assistance

In 2017, Zhang et al. designed a dynamic neck brace to measure and assist head motion in human users. The device offered accurate measurements of head motion and had the potential to improve the neck's range of motion for patients with head/neck posture disorders [36].

In 2018, Zhang et al. developed an active neck brace with three degrees of freedom designed to support patients exhibiting symptoms of head drooping. The device allowed for an enhanced range of motion for both the head and neck and reduced muscle activation when assisted by the brace [37].

In 2019, Liu et al. proposed a rigid–flexible parallel mechanism known as the 3-RXS, designed as a neck brace for patients suffering from head drooping symptoms. The mechanism demonstrated excellent rotational performance and could effectively assist in neck flexion, extension, lateral bending, and axial torsion. As shown in Figure 4b, the device was intended for use in various neurological disorders like amyotrophic lateral sclerosis (ALS), Parkinson's disease (PD), and primary lateral sclerosis (PLS), all of which may lead to head drooping syndrome (HDS). The 3-RXS mechanism facilitated passive neck extension, aiding in the correction of head drooping. The mechanism consisted of a series of links connected to both the top and fixed platforms through a sequence of joints in the following order: revolute joint (R), X-shaped compliant joint (X), and spherical joint (S). This construction was simple and lightweight, enabling smooth flexion/extension, left/right lateral bending, and axial rotational movements [38].

### 4.2.3. Parallel Robot for Neck Humanoid

In 2013, Gao et al. introduced a cable-driven flexible parallel robot designed to mimic the pitch and roll movements of the human neck. This robot employed three cables and a compression spring to serve as its flexible spine, replicating the motion of the head. Inverse kinematics were addressed using quaternion methods, and workspace analysis was performed under positive cable tension constraints, all validated through simulations [39].

In 2014, Gao et al. compared two different lateral bending models for the compression spring used in their cable-driven parallel robot. The robot aimed to emulate human neck movements by utilizing the spring's bending motion for inverse kinematics [40]. In a related study, Gao et al. presented a cable-driven flexible parallel robot designed to simulate the

motion of a human neck while minimizing motion noise. This robot used three cables and a compression spring, with the spring acting as the cervical spine to support the head-like moving platform and cables functioning as muscles around the human neck. Due to the flexible nature of the compression spring, inverse kinematics were not directly solvable. Cable placements were optimized to reduce actuation force, and workspace analysis was conducted under the constraint of positive cable tension. Simulations validated the efficacy of the inverse kinematics and workspace analysis [41].

In 2015, Jiang et al. presented a cable-driven flexible parallel robot featuring a compression spring as the cervical spine. As with previous models, the flexible nature of the spring meant that the inverse kinematics were not directly solvable. To find possible solutions, statical analysis was incorporated. The cable placements were optimized to reduce actuation force, and workspace analysis was conducted under a positive cable tension constraint [42].

In 2017, Gao et al. introduced another cable-driven parallel robot, this one featuring a flexible spine and four cables designed to mimic human neck movements. The lateral bending motion of the spring facilitated pitch and roll movements, while an included bearing enabled yaw motion. Both inverse kinematics and cable placement optimization were investigated through simulations [43].

In 2021, Quevedo et al. applied various linear and non-linear models to design a soft neck mechanism with a central soft link actuated by three motor-driven tendons. The force exerted on the individual tendons allowed the neck to perform motions similar to those of a human neck. The cable-driven parallel mechanism, referred to as 3-CDPM and illustrated in Figure 4c, was constructed from flexible materials and activated by cables, causing the upper platform to tilt. The neck was comprised of a base, movable platform, central soft link, tendons (cables), and motors [3].

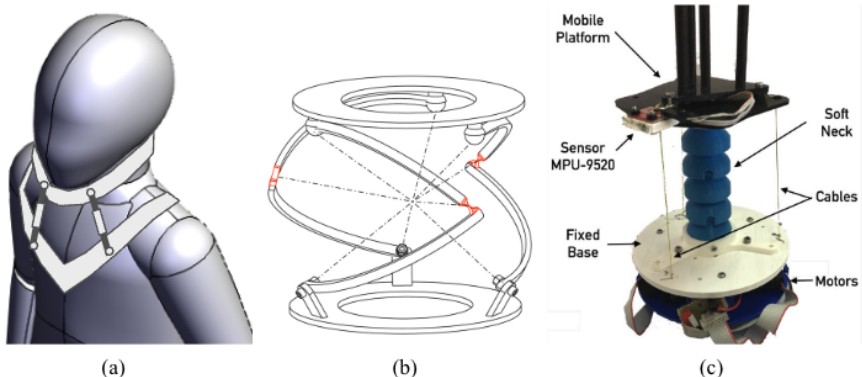

(a)        (b)        (c)

**Figure 4.** Neck joint mechanisms: (**a**) 3-RPS mechanism by Vellore Institute of Technology, redrawn based on [1]; (**b**) 3-RXS mechanism by the University of Technology Beijing [38], available under a Creative Commons Attribution License; (**c**) 3-CDPM mechanism by University Carlos III of Madrid [3], available under a Creative Commons Attribution License.

**Table 2.** Parallel robots for rehabilitation, assistance, and humanoids of the neck joint.

| Author | Year | Country | Device | TRL | Mechanism | DoF | ToM | Actuator | Model | Tool |
|---|---|---|---|---|---|---|---|---|---|---|
| Gao et al. [39] | 2013 | China | NH | 1 | cable-driven | 3 | PR | NS | IK | M |
| Gao et al. [40] | 2014 | China | NH | 2 | cable-driven | 3 | NS | NS | IK | M |
| Gao et al. [41] | 2014 | China | NH | 1 | cable-driven | 3 | NS | NS | IK | M |
| Jiang et al. [42] | 2015 | China | NH | 2 | cable-driven | 3 | NS | NS | IK | M |
| Gao et al. [43] | 2017 | China | NH | 2 | cable-driven | 3 | PR | NS | IK | M |
| Zhang et al. [36] | 2017 | USA | NA | 3 | 3-RRS | 3 | FE, RLB, RLR | NS | FK | M |
| Zhang et al. [37] | 2018 | USA | NA | 3 | 3-RRR | 3 | FE, RLB, RLR | ER | IK | NS |
| Liu et al. [38] | 2019 | China | NA | 3 | 3-RXS | 3 | FE, RLB, RLR | NS | IK | AN |

**Table 2.** *Cont.*

| Author | Year | Country | Device | TRL | Mechanisms | DoF | ToM | Actuator | Model | Tool |
|---|---|---|---|---|---|---|---|---|---|---|
| Lingampally et al. [1] | 2019 | India | NR | 3 | 3-RPS | 3 | RLR | EL | IK | M |
| Zhang et al. [2] | 2019 | USA | NR | 3 | 3-RRR | 3 | FE, RLB, RLR | ER | D | M |
| Quevedo et al. [3] | 2021 | Spain | NH | 3 | cable-driven | 3 | FE, RLR | ER | IK | M |
| Lozano et al. [4] | 2022 | Mexico | NR | 4 | 4-SPS | 4 | NS | EL | D | M |
| Zhang et al. [35] | 2023 | China | NR | 4 | cable-driven | 3 | FE, RLB, RLR | NS | IK | M |

Abbreviations: NH—neck humanoid, NA—neck assistance, NR—neck rehabilitation, 3-RRS—3 (revolute-revolute-spherical), 3-RRR—3 (revolute-revolute-revolute), 3-RXS—3 (revolute-joint X-spherical), 3-RPS—3 (revolute-prismatic-spherical), 4-SPS—4 (spherical-prismatic-spherical), PR—pitch and roll, NS—not specified, FE—flexion–extension, RLB—right–left bending, RLR—right–left rotation, ER—electric rotary, EL—electric linear IK—inverse kinematic, FK—forward kinematic, D—dynamics, M—MATLAB, AN—ANSYS.

### 4.3. Parallel Robot for Shoulder Joint

Studies related to rehabilitation, assistance, and humanoid applications for the shoulder joint are summarized in Table 3.

#### 4.3.1. Parallel Robot for Shoulder Rehabilitation

In 2014, Klein et al. introduced a novel robotic interface to explore the neuromechanical control of redundant planar arm movements. This device featured a 5R closed-loop pantograph design with a wrist flexion/extension cable-actuated mechanism. The interface's characteristics, such as motion range, impedance, friction, and dynamics, were discussed. This lightweight, high-force, and low-impedance device enabled research into redundant motor control in humans [44].

In 2015, Enferadi et al. proposed a new spherical parallel robot design for rehabilitation applications. The robot's full rotational capabilities were highlighted. Dimensional optimization, aimed at maximizing the robot's workspace, was carried out using genetic algorithms. The robot boasted a relatively large workspace and exhibited precision in its kinematics, Jacobian matrices, and workspace analysis [45].

In 2016, Enferadi et al. presented another spherical parallel robot designed for various applications, including rehabilitation (e.g., TV satellite dishes; tracking systems; solar panels; cameras; telescopes; and the rehabilitation of human joints like the ankle, shoulder, and wrist). The robot allowed for complete rotation around an axis. A genetic algorithm optimized its dimensions to maximize the workspace, which was found to be relatively large and free from singularities—a significant advantage [46]. Hunt et al. introduced a low-inertia shoulder exoskeleton with five degrees of freedom (DoF). The first innovation involved a 3DoF spherical parallel manipulator (SPM), which was designed through a new approach that mechanically coupled certain degrees of freedom to constrain the kinematics. The second innovation was a 2DoF passive slip interface that enhanced system mobility and prevented joint misalignment caused by the user's glenohumeral joint motion. Motion capture validated the SPM's kinematics, confirming both its forward and inverse kinematic solutions. Beyond its application for shoulder rehabilitation, the device introduced a novel motion coupling method that was applicable to various parallel architectures. It also showcased the versatility of its passive slip interface in both parallel and serial robotic systems [47].

In 2020, Niketkaliyev et al. shifted the focus to robotic shoulder rehabilitation exoskeletons, which often neglect certain shoulder DoF, leading to discomfort due to joint axis misalignments. They introduced a bio-inspired 5DoF hybrid human–robot mechanism (HRM) that combined serial and parallel manipulators with rigid and cable links. This hybrid mechanism ensured compatibility between human and exoskeleton joint axes and covered the complete range of human shoulder motion in a singularity-free workspace. Numerical simulations and a 3D-printed prototype validated the kinematic model and advantages of the proposed hybrid mechanism [48].

In 2021, Hunt proposed a novel parallel-actuated exoskeleton architecture for rehabilitation. This architecture's stiffness property could be optimized for specific tasks using a

stiffness model and bounded nonlinear multi-objective optimization. Figure 5a shows an exoskeleton designed with a four-bar system in a parallel mechanism, denoted as 4B-SPM. This system allowed individuals to modulate their stiffness attribute to optimize activities, such as augmented lifting or impact absorption for the shoulder [49].

### 4.3.2. Parallel Robot for Shoulder Assistance

In 2013, Sekine et al. presented a systematic approach to designing a shoulder prosthesis with consideration for force and spatial accessibility. Using measurements from Activities of Daily Living (ADLs), the design process entailed evaluating both force and spatial accessibility, followed by optimization based on kinematic and static models. The optimized parallel mechanism was tailored for specific ADL tasks and various spatial specifications, illustrating the potential for individualized shoulder prosthesis design. Figure 5b depicts an optimized, compact, pneumatic-actuator-driven parallel mechanism for a shoulder prosthetic arm [50].

In 2015, Sekine et al. proposed shoulder prostheses designed for transhumeral and shoulder disarticulation amputees. Accessibility and intrinsic viscoelasticity were the focal points of this study. The paper introduced new mechanisms—specifically, an antagonistic mechanism and a soft backbone—to enhance spatial characteristics and responsiveness to disturbances. The evaluation confirmed increased workspace and disturbance responsiveness for the prosthetic arm [20].

In 2018, Leal-Naranjo et al. introduced a synthesis of a spherical parallel manipulator for a seven-degrees-of-freedom (7DoF) prosthetic human arm. The design objectives included workspace, dexterity, and actuator torques. Optimization was performed using genetic algorithms, culminating in a manipulator that met all performance requirements [51]. Hunt et al. unveiled a new parallel-actuated exoskeleton architecture aimed at offering a superior alternative to serial actuation for augmenting multiple-DoF biological joints. This architecture employed a spherical parallel manipulator (SPM) with three four-bar substructures to control three rotational DoF independently. Variants of the four-bar spherical parallel manipulator (4B-SPM) were presented for shoulder, hip, wrist, and ankle exoskeletons. Three actuation methods for the 4B-SPM were explored, each based on different dynamic performance requirements. This work set the stage for advancements in more effective parallel-actuated exoskeletons as opposed to conventional serial-actuated counterparts [52]. Leal-Naranjo presented a low-cost prosthetic device designed for shoulder disarticulation and featuring seven DoF. The mechanisms for shoulder, elbow, and wrist movements were discussed. Dynamic simulations and experimental evaluations confirmed the device's functionality and suitability for daily activities [19]. Figure 5c illustrates a shoulder mechanism integrated into an arm prosthesis. This spherical mechanism, configured as a 3-RRR type, allowed for three degrees of freedom with shoulder motions, using compact motors. Given that the shoulder prosthetic supported the entire structure of the device, this is where the highest joint loads developed [19].

### 4.3.3. Parallel Robot for Shoulder Humanoid

In 2012, Chen et al. presented a novel homing algorithm for a three-degrees-of-freedom (3DoF) parallel spherical joint in a cable-driven parallel robot. The algorithm utilized incremental encoders and limit switches to identify the home posture automatically, implementing decoupling control for each axis. Simulation results affirmed the algorithm's effectiveness [53]. Both control accuracy and a consistent initial posture are pivotal when evaluating control algorithms. To minimize cumulative errors during control processes or to estimate initial postures, robotic systems need to revert to an approximate home posture.

In 2013, Wang et al. aimed to improve the performance of bionic joints using a five-link parallel mechanism, actuated by two antagonistic artificial pneumatic muscles (PMs). The study examined kinematics, singularity constraints, and joint torque models based on spring-damp dynamics. The joint's compliance, represented by the angle-to-spring torque ratio, was derived. Energy consumption analysis was conducted considering varying PM

lengths. The proposed bionic shoulder and elbow joints exhibited an enhanced angular range and decreased PM contraction, contributing to a more humanoid-like design [54].

In 2016, Alfayad et al. undertook a research initiative focused on developing a three-degrees-of-freedom (3DoF) hybrid mechanism suitable for humanoid robotics. This mechanism catered to various modules in humanoid robots and could also be adapted for other legged robots, such as quadrupeds and hexapods. Utilizing kinematic synthesis, the study proposed a novel hip mechanism that combined one rotary and two linear actuators. This approach accommodated the wide motion ranges of the shoulder module and introduced a new perspective on the contributions of linear actuators in both motion and force generation. The research employed kinematic and geometrical models to optimize the hybrid mechanisms, demonstrating their broad applicability in various robotic systems [55].

In 2017, Jiang et al. focused on developing a new hybrid mechanism for humanoid wrist and shoulder joints. A cable-driven parallel robot platform was developed for experimental study. A dynamic model of the mechanism was formulated, considering the coupling theory of flexible body motion and deformation. A nonlinear control method was applied for anti-vibration control. Both simulations and experimental results validated the feasibility and control scheme of the hybrid mechanism [56].

In 2019, Liu et al. introduced a bionic flexible manipulator driven by pneumatic muscle actuators (PMAs). The study outlined the configuration design based on human physiological mapping, established kinematic models, and employed the Lagrange method for dynamic analysis. A fuzzy torque control algorithm, developed using the computed torque method, showed improved trajectory tracking and accuracy compared to traditional methods [57]. In the same year, Bai et al. provided a comprehensive review of state-of-the-art techniques in spherical motion generation via parallel manipulators or spherical motors. The review covered kinematics, dynamics, design optimization, and emerging applications, offering insights into new research challenges and future developments in the field [58].

In 2021, Wang et al. proposed a multi-objective trajectory planning approach for a 7DoF hybrid humanoid robotic arm. The methodology combined kinematic modeling and optimization to achieve faster transit times, lower energy consumption, and higher stability during point-to-point tasks. Simulation results validated the proposed approach [59].

In 2023, Chen et al. introduced an enhanced design and modeling analysis for a 3DoF series-parallel joint module used in humanoid service robots. This module, inspired by human-like 3DoF joints, employed a cable-driven method. Using the shoulder joint as a model, it could execute various motions like arm abduction, back extension, and lifting. The team designed a shared modular connector for easier assembly and disassembly across modules and a tension amplification mechanism for a more compact design. They also developed a transient torque model and a cross-coupling control framework and conducted kinematics analysis based on the anti-parallelogram principle. Prototyping and tests showed the design's potential for realistic humanoid shoulder movements, offering a novel concept for humanoid robots [60].

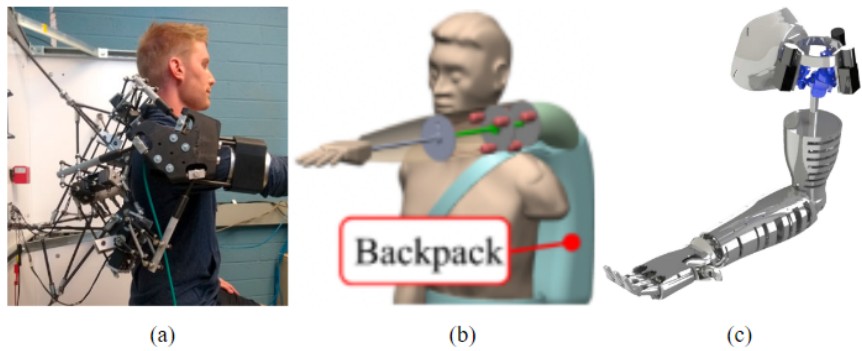

**Figure 5.** Shoulder joint mechanisms: (**a**) 4B-SPM mechanism by Arizona State University [49]; (**b**) cable-driven mechanism by Chiba University [50]; (**c**) 3-RRR mechanism by Universidad de Guanajuato [19]. All are available under a Creative Commons Attribution License.

**Table 3.** Parallel robots for rehabilitation, assistance, and humanoids of the shoulder joint.

| Author | Year | Country | Device | TRL | Mechanism | DoF | ToM | Actuator | Model | Tool |
|---|---|---|---|---|---|---|---|---|---|---|
| Chen et al. [53] | 2012 | China | SH | 2 | cable-driven | 7 | NS | ER | IK | AD |
| Sekine et al. [50] | 2013 | Japan | SA | 2 | cable-driven | 3 | FE, AA, IER | PL | FK | ADLA |
| Wang et al. [54] | 2013 | China | SH | 2 | 5-PMA | 5 | NS | P | FK | MS |
| Klein et al. [44] | 2014 | UK | SR | 4 | 5R | 3 | NS | E | FK | CAD |
| Sekine et al. [20] | 2015 | Japan | SA | 3 | 3-SPS/P | 3 | NS | PL | FIK | NS |
| Enferadi et al. [45] | 2015 | Iran | SR | 2 | 3-RSS/S | 3 | RP | ER | IK | NS |
| Enferadi et al. [46] | 2016 | Iran | SR | 2 | 3-RSS/S | 3 | NS | ER | IK | NS |
| Hunt et al. [47] | 2016 | USA | SR | 3 | SPM | 3 | NS | EL | FIK | CAD |
| Alfayad et al. [55] | 2016 | France | SH | 3 | 2-UPUR/RU | 2 | PRY | EL | IK | NS |
| Jiang et al. [56] | 2017 | China | SH | 3 | cable-driven | 2 | NS | ER | NS | NS |
| Leal-Naranjo et al. [51] | 2018 | Italy | SA | 3 | 3-RRR | 3 | FE | ER | IK | NS |
| Leal-Naranjo et al. [19] | 2018 | Mexico | SA | 2 | 3-RRR | 3 | FE | ER | IK | AD |
| Hunt et al. [52] | 2018 | USA | SA | 3 | 4B-SPM | 3 | NS | ER | FIK | M |
| Lui et al. [57] | 2019 | China | SH | 3 | 5-PMA | 5 | NS | PL | FIK | M-AD |
| Bai et al. [58] | 2019 | Denmark | SH | 3 | 3-RRR | 3 | RP | ER | FIK | NS |
| Niyetkaliyev et al. [48] | 2020 | Australia | SR | 2 | cable-driven | 3 | AA | NS | IK | CAD |
| Hunt et al. [49] | 2021 | USA | SR | 3 | 4B-SPM | 7 | FE, AA | EL | FIK | NS |
| Wang et al. [59] | 2021 | China | SH | 2 | 5R | 3 | FE, AA, IER | ER | FK | AD |
| Chen et al. [60] | 2023 | China | SH | 4 | cable-driven | 3 | FE, AA, L | ER | IK | NS |

Abbreviations: SR—shoulder rehabilitation, SH—shoulder humanoid, SA—shoulder assistance, 5-PMA—5 pneumatic muscle actuator, NS—not specified, 3-SPS/P—3 (spherical-prismatic-spherical)/1 (prismatic), 3-RSS/S—3 (revolute-spherical-spherical)/spherical, 3-RRR—3 (revolute-revolute-revolute), 4B-SPM—4-bar (spherical parallel manipulator), 5R—5 revolute, FE—flexion–extension, AA—abduction–adduction, IER—internal–external rotation, RP—rotational pure movement, L—lifting, PRY—pitch-roll-yaw, ER—electric rotary, EL—electric linear, PL—pneumatic linear, IK—inverse kinematic, FK—forward kinematic, FIK—forward and inverse kinematic, AD—ADAMS, ADLA—(ADL area) is used to evaluate spatial accessibility, MS—MATLAB-SimMechanics, CAD—Computer-Aided Design, M—MATLAB, M-AD—MATLAB-ADAMS.

### 4.4. Parallel Robot for Wrist Joint

Studies related to wrist assistance and rehabilitation and humanoid wrist mechanisms are summarized in Table 4.

#### 4.4.1. Parallel Robot for Wrist Rehabilitation

In 2013, Pehlivan et al. developed an adaptive controller for a robotic mechanism designed for the upper-extremity rehabilitation of the wrist. Compared to a proportional-derivative (PD) controller, the model-based adaptive controller improved trajectory tracking. The adaptive controller achieved similar error performance but used significantly lower feedback gains, offering a more compliant interface for patients during rehabilitation sessions [61].

In 2017, Bian et al. introduced an exoskeleton aimed at rehabilitating the elbow, forearm, and wrist motor functions in stroke patients. The EFW Exo II exoskeleton was based on a hybrid mechanism that combined a parallel 2-URR/RRS mechanism with a serial R mechanism. It included adjustable features to accommodate different arm sizes and utilized force sensors to facilitate patient interaction. The exoskeleton provided ranges of motion that met the requirements of activities of daily living [62].

In 2018, Kitano et al. unveiled a wearable wrist rehabilitation training device that utilized a parallel link mechanism. This innovative approach allowed for the training of both translational and rotational wrist joint motions, filling a gap left by previous techniques that did not address translational motion. The device could enable a range of motion that covered approximately 60% of the wrist's mobility, potentially reducing wrist joint strain [6].

In 2021, Wang et al. introduced a soft parallel robot designed for automated wrist rehabilitation. Figure 6a illustrates the 6-SPS/PS soft parallel robot that employed pneumatic artificial muscles for wrist rehabilitation. This robot merged soft and parallel structures to offer a secure, adaptable, and low-cost personalized rehabilitation solution. Linear actuators, including pneumatic artificial muscles, drove the robot, and an electromyogra-

phy sensor provided feedback for evaluating rehabilitation progress. Experimental tests confirmed the robot's efficacy in assisting various wrist motions [63].

In 2022, Goyal et al. developed an impedance controller designed to rehabilitate stroke patients with wrist motor impairments. The controller employed a Koopman-operator-based autodidactic system identification model to predict wrist joint stiffness during different rotational motions. It adjusted the applied force based on the subject's joint stiffness, utilizing a parallel-structured end-effector robot equipped with biomimetic muscle actuators. The performance of the controller was validated through tests on healthy subjects [64].

In 2023, Li et al. developed a bionic cable-driven mechanism for forearm-wrist rehabilitation. This device mimics human wrist motions with three degrees of freedom, addressing the full range of wrist and forearm actions. Notably, a spring within a parallel mechanism helps counteract cable slack. The system's design incorporates kinematics and statics for precise movement calculations. Simulations and practical tests confirmed its effectiveness and accuracy for rehabilitation purposes [65]. In the same year, Goyal et al. introduced a robot with a four-link parallel end-effector designed for wrist joint rehabilitation. Using biomimetic muscle actuators (BMAs), the robot had an inherent flexibility, with a fuzzy model developed to recognize the BMAs' nonlinear characteristics. The system's stiffness-observer tailored itself to individual subject stiffness, adjusting the robot's trajectories. An adaptive controller, based on the fuzzy model and stiffness-observer, governed the four BMAs, giving the robot end-effector three rotational degrees of freedom. Initial tests with three healthy individuals confirmed the controller's effectiveness in guiding the robot while accounting for the compliant and nonlinear nature of the BMAs, even adjusting for higher-stiffness areas within the wrist's range of motion [66]. In the same year, Goyal et al. developed a trajectory-tracking controller for a wrist rehabilitation robot with three degrees of freedom. They used the Koopman linear system to address the nonlinearity of human–robot interaction, turning state variables into linear approximations of nonlinear systems through Koopman operators. These operators, learned via linear regression, determined the system dynamics for the robot's trajectory controller. This data-driven method resulted in a clear control-oriented model. Tests with three healthy individuals proved the controller's effectiveness in guiding the wrist along a set trajectory [67].

4.4.2. Parallel Robot for Wrist Assistance

In 2012, Serracin et al. designed a parallel robot to assist in bone milling surgeries. The robot had two active degrees of freedom and was employed for orientation during bone-milling procedures. The paper outlined the kinematic geometry, discussed workspace optimization, and performed force analysis for jawbone reconstruction. The singularities of the mechanism were analyzed, and the motor selection was justified based on torque requirements. The study also presented simulation results and a prototype using linear motors [68].

In 2018, Lee et al. introduced an over-actuated coaxial spherical parallel mechanism optimized for efficient wrist motion. The mechanism's rotation axis was aligned with the wrist's pronation-supination movement. The prototype demonstrated a design that enhanced user comfort and workspace efficiency. Simulation results indicated improved performance compared to similar devices, along with extensive motion coverage [69]. Figure 6b depicts the wrist human–machine interface based on this over-actuated coaxial spherical parallel mechanism. The base was connected to two links that were coaxially coupled, and the rotation axis was specifically designed to align with the wrist's pronation/supination motion, which has the most extensive operational range of wrist movements [69].

In 2021, Lee et al. proposed a robotic exoskeleton interface (REI) for the wrist, based on a fully actuated coaxial spherical parallel mechanism (CSPM). The CSPM-based interface offered pure rotational motion akin to the human wrist, along with a high torque output

and low moving-parts inertia. The device's torque output and range of motion aligned closely with human capabilities [70].

In 2022, Lopez-Custodio et al. conducted a stiffness analysis of the Exechon hybrid manipulator, which served as a five-axis machine tool. The study included the consideration of an offset wrist in both kinematic and stiffness analyses. A compliance model was formulated and validated against experimental data, offering a more accurate portrayal of the manipulator's behavior [71]. Sanjuan et al. proposed a cable-driven wrist prosthesis (CDWP) to address challenges related to limb amputation. The anti-parallel-based local transmission index was introduced to optimize the device's dimensions. The CDWP design aimed to offer both high dexterity and aesthetic appeal to a wide range of patients [72].

### 4.4.3. Parallel Robot for Wrist Humanoid

In 2013, Chaparro-Altamirano et al. proposed a system utilizing a 3SPS-1S parallel manipulator for surveillance and defense applications. The mechanism incorporated a central leg to increase stiffness and provide three pure rotational degrees of freedom. The study covered inverse kinematics, forward kinematics solved through geometry and neural networks, workspace calculations, parameter optimization, and singularity detection via Jacobian matrix analysis [73].

In 2015, Kong et al. focused on synthesizing two-degrees-of-freedom (DoF) parallel mechanisms (PMs) capable of both spherical translation and sphere-on-sphere rolling modes. They introduced a 2DoF 3-4R overconstrained PM, derived from an existing 5DoF US equivalent PM, to serve as the basis for further developments. The study classified 2DoF 3-4R PMs capable of both identified modes by exploring shared conditions between them [74].

In 2017, Lu et al. conducted a comprehensive study analyzing coordinated grasping kinematics for multi-fingered systems and optimizing grasping force in a parallel hybrid hand. The study included conditions for coordinated grasping and formulas for calculating the displacement, velocity, and acceleration of contact points between the fingers and the object [75].

In 2018, Wu et al. addressed the dynamic stability challenges of a tripod parallel robotic wrist using the monodromy matrix method. The wrist demonstrated uninterrupted end-effector rotation across its orientation workspace, making it suitable for machine tool head tasks like drill point grinding. Stability analysis was conducted using Floquet theory, and stability charts were constructed to identify parametric instabilities [76].

In 2019, He et al. introduced a novel concept of a self-insulating joint design, employing a cable-driven parallel-series (CDPS) mechanism coupled with electrical insulation analysis. This design offered mechanical support and electrical insulation during live-line operations, thus facilitating equipment maintenance without manual intervention or power interruption. Figure 6c depicts the wrist joint of the robotic arm as a CDPS mechanism, featuring a lower and an upper platform in concentric positions for mechanical support. Two concentric shafts in a spring, connected by a universal joint, provided additional structural support. Four cables powered the structure [77].

In 2020, Pang et al. proposed a bionic flexible wrist parallel mechanism that mimicked human wrist joints by using a combination of rope-driven and compression spring-supported hybrid mechanisms. A parallel structure controlled by cables emulated wrist muscles. The inverse kinematics were solved using force and torque balance conditions along with the spring bending equation. MATLAB software was used for analysis, verifying the mechanism's feasibility for wrist rehabilitation and promoting the development of rehabilitation robots and rope-driven technology [9].

In 2021, Wang et al. proposed a seven-degrees-of-freedom serial–parallel hybrid humanoid robotic arm. The arm's trajectory planning was optimized for speed, energy consumption, and stability using a multi-objective optimization approach. Simulation results validated the effectiveness of this methodology [59].

In 2022, Bazman et al. introduced a 4DoF parallel forceps mechanism designed for minimally invasive surgery. This mechanism employed a unique center leg to convert thrust

into gripping motion. The study included kinematic analysis and workspace assessment and addressed unintended gripper motion. Human-in-the-loop simulations validated the design [78]. Li et al. developed a 3-RPS/US parallel mechanism with two DoF to enhance load-bearing capabilities. The study covered kinematics, Jacobian matrix analysis, workspace calculations, singularity considerations, and static analysis. The potential application of this mechanism in tracking photovoltaic brackets was demonstrated [79].

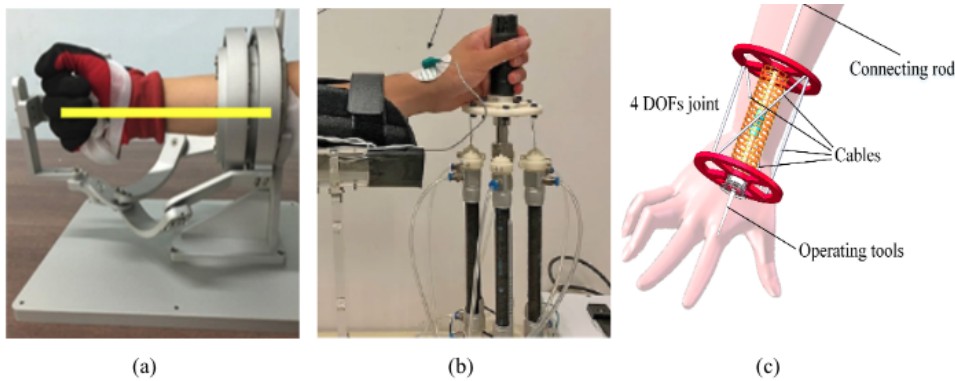

**Figure 6.** Wrist joint mechanisms: (**a**) 6-SPS/PS mechanism by the University of Macau [63]; (**b**) 3-CSPM mechanism by Kwangwoon University [69]; (**c**) 4-CDPS mechanism by Southeast University [77]. All are available under a Creative Commons Attribution License.

**Table 4.** Parallel robots for rehabilitation, assistance, and humanoids of the wrist joint.

| Author | Year | Country | Device | TRL | Mechanism | DoF | ToM | Actuator | Model | Tool |
|---|---|---|---|---|---|---|---|---|---|---|
| Serracin et al. [68] | 2012 | Spain | WA | 3 | 2-UPS/S | 2 | NS | EL | IK | M |
| Chaparro et al. [73] | 2013 | Mexico | WH | 3 | 3-SPS/S | 3 | RPY | EL | FK | NS |
| Pehlivan et al. [61] | 2013 | USA | WR | 3 | 3-RPS | 2 | FE, AA | PL | FIK | M |
| Kong et al. [74] | 2015 | UK | WH | 2 | 3-4R | 2 | NS | NS | IK | NS |
| Lu et al. [75] | 2017 | China | WH | 3 | 5-SPM | 5 | NS | EL | FIK | NS |
| Bian et al. [62] | 2017 | China | WR | 3 | 2-URR/RRS | 3 | FE, AA | ER | IK | NS |
| Lee et al. [69] | 2018 | South Korea | WA | 3 | 2-RRR | 3 | FE, AA | ER | FIK | NS |
| Kitano et al. [6] | 2018 | Japan | WR | 3 | 3-RRR | 3 | FE, AA, SP | ER | FIK | NS |
| Wu et al. [76] | 2018 | China | WH | 3 | 2-PUU/RPS | 2 | TR | ER, EL | IK | NS |
| He et al. [77] | 2019 | China | WH | 2 | cable-driven | 4 | NS | ER | IK | NS |
| Pang et al. [9] | 2020 | China | WH | 3 | cable-driven | 2 | FE, AA | ER | IK | NS |
| Wang et al. [63] | 2021 | China | WR | 3 | 6-SPS/PS | 3 | FE, AA, SP | PL | FIK | NS |
| Lee et al. [70] | 2021 | Korea | WA | 3 | 2-CSPM | 2 | FE, AA, SP | EL | FIK | NS |
| Wang et al. [59] | 2021 | China | WH | 2 | 3-UPS/S | 2 | FE, SP | EL | FK | AD |
| López et al. [71] | 2022 | UK | WA | 3 | 2-RRRPR/RRPR | 3 | NS | EL | IK | M |
| Goyal et al. [64] | 2022 | Australia | WR | 3 | 3-RPR | 3 | FE, AA, PS | PMA | IK | NS |
| Bazman et al. [78] | 2022 | Turkey | WH | 2 | 3-RSR/UUP | 2 | RPY | NS | FK | MS |
| Sanjuan et al. [72] | 2022 | USA | WA | 2 | cable-driven | 2 | NS | NS | IK | NS |
| Li et al. [79] | 2022 | China | WH | 2 | 3-RPS/US | 2 | NS | EL | IK | NS |
| Li et al. [65] | 2023 | China | WR | 4 | cable-driven | 3 | FE, AA, SP | ER | IK | M |
| Goyal et al. [66] | 2023 | Australia | WR | 4 | 4-BMA | 3 | FE, AA, SP | PL | IK | NS |
| Goyal al. [67] | 2023 | Australia | WR | 4 | 4-BMA | 3 | FE, AA, SP | PL | IK | NS |

Abbreviations: WR—wrist rehabilitation, WA—wrist assistance, WH—wrist humanoid, 2-UPS/S—2 (universal-prismatic-spherical)/1 (spherical), 3-SPS/S—3 (spherical-prismatic-spherical)/1 (spherical), 3-RPS—3 (revolute-prismatic-spherical), 3-4R—3 (4 revolute), 5-SPM—5 (spherical parallel mechanism), 2-URR/RRS—2 (universal-revolute-revolute)/1 (revolute-revolute-spherical), 2-RRR—2 (revolute-revolute-revolute), 3-RRR—3 (revolute-revolute-revolute), 2-PUU/RPS—2 (prismatic-universal-universal)/1 (revolute-prismatic-spherical), 6-SPS/PS—6 (spherical-prismatic-spherical)/1 (prismatic-spherical), 2-CSPM—2 (coaxial spherical parallel mechanism), 3-UPS/S—3 (universal-prismatic-spherical)/1 (spherical), 2-RRRPR/RRPR—2 (revolute-revolute-revolute-prismatic-revolute)/1 (revolute-revolute-prismatic-revolute), 3-RPR—3 (revolute-prismatic-revolute), 3-RSR/UUP—3 (revolute-spherical-revolute)/1 (universal-universal-prismatic), 3-RPR—3 (revolute-prismatic-revolute), 3-UPU—3 (universal-prismatic-universal), 3-RPS/US—3 (revolute-prismatic-spherical)/3 (universal-spherical), 4-BMA—4 (biomimetic actuators) RPY—roll-pitch-yaw, FE—flexion–extension, AA—abduction–adduction, SP—supination–pronation, TRP—translation-rotation-pure, EL—electric linear, PL—pneumatic linear, ER—electric rotary, IK—inverse kinematic, FK—forward kinematic, FIK—forward and inverse kinematic, NS—not specified, M—MATLAB, AD—ADAMS, MS—MATLAB-SimMechanics.

### 4.5. Parallel Robot for Hip Joint

Studies related to assistance, rehabilitation, and humanoid applications for the hip are summarized in Table 5.

### 4.5.1. Parallel Robot for Hip Rehabilitation

In 2016, Rastegarpanah et al. investigated a six-degrees-of-freedom parallel robot developed at the University of Birmingham for use in hip rehabilitation for stroke patients. This robot could replicate foot trajectories associated with three types of rehabilitation exercises: hip flexion/extension, ankle dorsiflexion/plantarflexion, and human gait. The study also emphasized the robot's ability to lift significant loads, indicating its potential for the effective administration of lower-limb rehabilitation exercises [80].

In 2020, Zhang et al. introduced an innovative approach to enhance the performance of a three-degrees-of-freedom parallel mechanism used in hip rehabilitation robots. The objective was to prevent kinematic singularity within the workspace corresponding to human gait and to improve power efficiency. This was achieved by optimizing the geometric parameters of the mechanism and by proposing improved force transmission indices. Optimization was conducted using a multi-objective model and the differential evolution algorithm. The effectiveness of this method was validated by comparing the performance metrics of the optimized mechanism to those of the original model, particularly in terms of Jacobian matrix singularity and output power efficiency [81].

In 2022, Shi et al. explored a serial–parallel lower-limb rehabilitation exoskeleton for its potential to mimic human lower-limb kinematics. To address challenges related to modeling errors and uncertainties, an adaptive control approach was proposed. This approach incorporated sensor data from the parallel mechanism to accurately capture attitude and employed a neural network adaptive controller to compensate for uncertainties and external disturbances. The effectiveness of the dynamic modeling and control system was validated through experimental testing [82].

### 4.5.2. Parallel Robot for Hip Assistance

In 2019, Ren et al. designed a wearable lower-limb exoskeleton to assist with medical rehabilitation. Traditional exoskeletons often struggle to precisely replicate natural human limb movements. To address this issue, the researchers developed an innovative anthropomorphic lower-limb exoskeleton based on a series-parallel mechanism. Human lower-limb movements were captured using an optical gait-tracking system. By comparing the simulation results from the series-parallel mechanism with the captured human data, a kinematics matching model was established specifically for the hip joint. The study demonstrated that the proposed model effectively reduced kinematic matching errors in multiple directions. This made the anthropomorphic series-parallel mechanism a significant improvement for tracking human hip joint positions. Figure 7a shows a lower-limb exoskeleton based on the series-parallel mechanism. A 6-SPS parallel mechanism was employed to enhance the pelvis's capacity to move, thereby providing the lumbar spine joint with six degrees of freedom (DoF) [83].

In 2020, Song et al. focused on aiding amputees in regaining their daily quality of life through the development of a hip prosthetic mechanism, as shown in Figure 7b. Analyzing human hip motion characteristics, they designed a 2-UPR/URR parallel mechanism with a passive limb. The study delved into the inverse kinematics of this mechanism using a closed-loop vector method. A comprehensive analysis of constrained and kinematic Jacobian matrices was conducted, leading to the construction of a $6 \times 6$ fully populated Jacobian matrix. This matrix aided in evaluating kinematic performance. Additionally, a dynamic model based on the virtual work principle was formulated. Its theoretical solution was compared with numerical simulation results, validating the dynamic model's efficacy and the accuracy of the inverse kinematics. The prosthetic mechanism exhibited a larger rotating workspace and superior mechanical performance, closely mimicking the range of motion and bearing capacity of the human hip across various gait modes. The torque

change during hip flexion and extension aligned well with human hip behavior, affirming the feasibility and dynamic performance of the prosthetic hip mechanism [84].

In 2023, Wang et al. introduced a hip exoskeleton with a distinctive parallel structure that allowed unrestricted walking and resolved misalignment issues. They proposed a model-based controller rooted in a human–machine integrated dynamic model, enhancing the system's responsiveness to user movements. Unlike most existing exoskeletons, this controller only needed the user's kinematic data, not interaction force, making the system more compact. Tests showcased the proposed hip exoskeleton's kinematic compatibility and assistance capabilities [85].

### 4.5.3. Parallel Robot for Hip Humanoid

In 2017, Jiang et al. addressed challenges related to mimicking the distribution of human muscles in robots actuated by antagonistic pneumatic artificial muscles. They observed that existing control algorithms often fail to replicate natural muscle patterns accurately. To remedy this, the researchers proposed a humanoid lower-limb parallel mechanism with unevenly distributed muscle representation. They analyzed the kinematics and dynamic movements of the bionic hip joint using an observer-based fuzzy adaptive control algorithm. This algorithm considered the overall movement of the joint rather than focusing on individual pneumatic artificial muscles. The parameters were optimized using a neural network, and experimental results confirmed the effectiveness of this method. The study particularly highlighted the importance of muscle roles in trajectory tracking for specific muscle groups [22]. Wang et al. focused on the dynamic performance of a 4-SPS/CU parallel mechanism that included a spherical joint with clearance. They employed Archard's wear model to analyze the wear properties of the socket. A kinematics model for the spherical joint with clearance was established, and an improved contact force model was introduced. The researchers formulated a dynamic model for the parallel mechanism while taking into account the spherical joint with clearance. Wear analysis involved the decomposition of contact forces and the computation of sliding distances. Simulation results revealed a nonuniform wear depth along the socket surface, which had implications for the mechanism's performance [86].

In 2018, Russo et al. presented the mechanical design of LARMbot 2, an affordable humanoid robot intended for user-centric applications. LARMbot 2 featured parallel architectures for both the torso and legs, as shown in Figure 7c. The design's kinematics for its primary components—including legs, torso, and arms—are detailed and compared with the previous version. A prototype was introduced, showcasing the subsystem construction and technical specifications. Experimental results offered insights into the robot's performance during walking and weight-lifting operations [87].

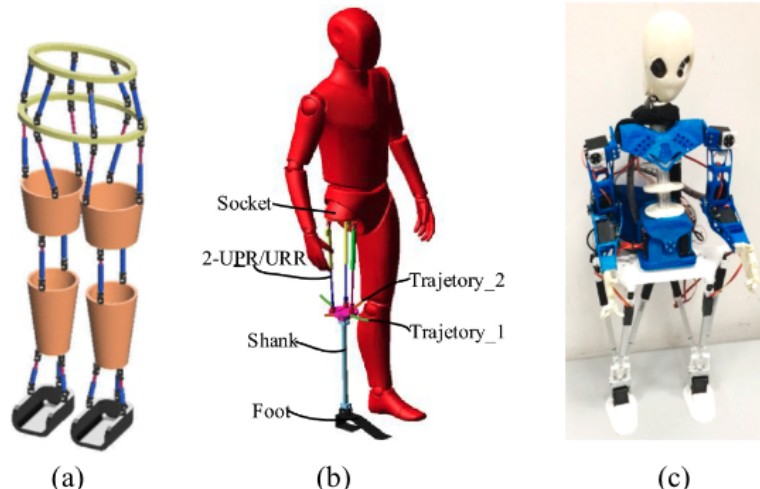

(a)　　　　　(b)　　　　　(c)

**Figure 7.** Hip joint mechanisms: (**a**) 6 SPS mechanism by Shangai University [83]; (**b**)4 RPS mechanism by Jiaotong University [84]; (**c**)3UPR mechanism by the University of Cassino and Southern Latium [87]. All are available under a Creative Commons Attribution License.

**Table 5.** Parallel robots for rehabilitation, assistance, and humanoids of the hip joint.

| Author | Year | Country | Device | TRL | Mechanism | DoF | ToM | Actuator | Model | Tool |
|---|---|---|---|---|---|---|---|---|---|---|
| Rastegarpanah et al. [80] | 2016 | UK | HR | 4 | 6-UPS | 6 | FE | EL | FK | M |
| Jiang et al. [22] | 2017 | China | HH | 3 | PMA | NS | NS | PL | FIK | NS |
| Wang et al. [86] | 2017 | China | HH | 2 | 4-SPS/CU | 2 | NS | EL | NS | NS |
| Russo et al. [87] | 2018 | Italy | HH | 3 | 4-SPS | 4 | NS | EL | IK | NS |
| Ren et al. [83] | 2019 | China | HA | 3 | 6-SPS | 6 | NS | PL | FIK | NS |
| Zhang et al. [81] | 2020 | China | HR | 3 | 2-UPS/RRR | 3 | FE, AA, IER | EL | FK | NS |
| Song et al. [84] | 2020 | China | HA | 2 | 4-RPS | 3 | NS | NS | FK | NS |
| Shi et al. [82] | 2022 | China | HR | 3 | 2-UPS/RRR | 3 | FE-AA, IER | EL | FIK | NS |
| Wang et al. [85] | 2023 | China | HA | 4 | 2-UPS+S | 2 | IER | EL | IK | NS |

Abbreviations: HR—hip rehabilitation, HA—hip assistance, HH—hip humanoid, 6-UPS—6 (universal-prismatic-spherical), PMA—pneumatic muscle actuator, 4-SPS/CU—4 (spherical-prismatic-spherical)/1 (cylindrical-universal), 3-UPR—3 (universal-prismatic-revolute), 6-SPS—6 (spherical-prismatic-spherical), 2-UPS-RRR—2 (universal-prismatic-spherical)/1 (revolute-revolute-revolute), 4-RPS—4 (revolute-prismatic-spherical), 2-UPS/RRR—2 (universal-prismatic-spherical)/1 (revolute-revolute-revolute), FE—flexion–extension, NS—not specified, AA— abduction–adduction, IER—internal–external rotation, ER—electric rotary, EL—electric linear, PL—pneumatic linear, IK—inverse kinematic, FK—forward kinematic, FIK—forward and inverse kinematic, M—MATLAB.

*4.6. Parallel Robot for Ankle Joint*

In the realm of scientific exploration concerning parallel robots, it is paramount to emphasize the scarcity of articles related to assistance and humanoids in the current body of literature. Notably, existing studies tend to focus exclusively on ankle rehabilitation, leaving a significant gap in our understanding of the broader landscape of assistance and humanoid interactions.

Parallel ankle rehabilitation devices aim to improve strength and mobility in patients suffering from ankle injuries, such as sprains or fractures. These devices provide controlled resistance and enable precise ankle motion. Studies related to rehabilitation are summarized in Table 6, each of which is described below.

In 2012, Wang et al. introduced a novel parallel ankle rehabilitation robot and determined its efficacy through kinematic analysis and simulations. The results showcased the robot's versatility under different input scenarios. with single inputs, the robot facilitated a range of ankle movements including dorsiflexion (0–30°), plantar flexion (0–50°), inversion/eversion (0–18°), and adduction/abduction (0–10°), making it suitable for initial ankle rehabilitation training. For double inputs, the robot enabled ankle motions such as dorsiflexion (0–30°), plantar flexion (0–50°), inversion/eversion (0–25°), and adduction/abduction (0–20°), apt for medium-term rehabilitation training. Under three inputs, the robot supported dorsiflexion (0–30°), plantar flexion (0–50°), inversion/eversion (0–40°), and adduction/abduction (0–30°), optimally meeting the needs of comprehensive ankle rehabilitation training. The robot's design effectively accommodated the full range of motion necessary for normal ankle function and provided a versatile platform for various rehabilitation exercises, thereby aiding patients' recovery processes [88].

In 2013, Saglia et al. examined the control architecture and experimental outcomes of the Ankle Rehabilitation Robot (ARBOT), aiming to develop effective control algorithms for aiding ankle joint training and rehabilitation, particularly in the presence of musculoskeletal injuries. They utilized a position control approach for patient-passive exercises and an admittance control technique for patient-active exercises, both with and without motion assistance. The design of the control algorithms was informed by an analysis of the rehabilitation protocol, taking into account system dynamics and human–robot interaction. Experimental assessments involving healthy subjects were conducted to evaluate the performance of the proposed control algorithms [89]. In the same year, Wang et al. presented the design of a novel 3-RUS/RRR redundantly actuated parallel mechanism for ankle rehabilitation, based on conceptual design principles. The mechanism facilitated ankle rotational movements in three directions, aligning the mechanism's center of rotations with the ankle axes. A new actuator redundancy scheme was introduced to enhance

system flexibility without compromising inherent advantages. Kinematic attributes such as dexterity, singularity, and stiffness were evaluated using the Jacobian matrix, which was then followed by simulations [90].

In 2014, Jamwal et al. introduced an innovative wearable ankle robot designed for the physical rehabilitation of ankle sprains. Originating from a comprehensive analysis of existing ankle robots, this bioinspired design was adaptable to individuals across different physiological abilities and age groups. The robot was powered by lightweight yet robust pneumatic muscle actuators (PMAs) that emulated skeletal muscles. To address the PMAs' nonlinear characteristics, a fuzzy-based disturbance observer (FBDO) was employed. Additionally, an adaptive fuzzy logic controller—based on Mamdani inference and augmented with FBDO—managed the transient behavior of the PMAs. This control scheme allowed for the simultaneous control of four parallel actuators, achieving three rotational degrees of freedom. Experimental evaluations were conducted with a neurologically intact subject to maintain active–passive robot–human interaction during predefined trajectory tracking. These trials accounted for unforeseen human interventions, nonlinear and compliant actuators, and the parallel kinematic structure of the ankle robot [91].

In 2015, Jamwal et al. delved into the design, analysis, and optimization of a novel wearable parallel robot aimed at ankle rehabilitation. To confront challenges related to parallel mechanisms, flexible actuators, and ankle rehabilitation constraints, a thorough design analysis was executed. Three design stages—kinematic design, actuation design, and structural design—were meticulously investigated, resulting in six critical performance objectives essential for achieving the design goals. Initially, the optimization focused on single objectives; however, due to conflicting objectives, a simultaneous optimization approach was required. The study employed the nondominated sorting algorithm (NSGA II), based on evolutionary algorithms, for multi-objective optimization. NSGA II outperforms single-objective and preference-based optimization methods, providing more comprehensive design solutions. Furthermore, a fuzzy-based ranking method was introduced to select the ultimate design from NSGA II's set of nondominated solutions. This methodology is adaptable for various types of parallel robots [92]. In the same year, Valles et al. from Universitat Politécnica de Valencia developed an economical parallel rehabilitation robot, addressing its design, kinematics, dynamics, and control features. Various position and force controllers were examined to ensure precise tracking performance. The robot was equipped with a force-sensor-integrated orthopedic boot designed for ankle exercises targeting injured areas. It supported passive, active-assistive, and active-resistive exercises for dorsi/plantar flexion, inversion, and eversion ankle movements. Orocos, a component-based middleware, offered a modular and configurable control scheme. Integration with Orocos and ROS enabled real-time teleoperation, represented by a CAD model that mirrored the robot's position. Teleoperated rehabilitation exercises could be conducted using devices like a Wiimote [93].

In 2016, Jamwal et al. developed an ankle rehabilitation robot that employed an interactive training approach based on impedance control. Powered by PMAs, the robot allowed patients to adapt robot-induced motions to their specific limb movements, accommodating for disabilities. Four training modes—position control, zero-impedance control, non-zero-impedance control with high compliance, and non-zero-impedance control with low compliance—were employed to assess the effectiveness of the proposed control scheme. Evaluations involving 10 neurologically intact subjects indicated that increased robotic compliance led to greater participant engagement during training [94]. In the same year, Ruiz-Hidalgo et al. introduced a novel three-degrees-of-freedom (DoF) parallel robot that utilized revolute and spherical joints. The inverse kinematic model incorporate a PID-type controller with tracking capabilities to accurately follow a desired trajectory. Potential applications for this robot include its use as a motion simulator or ankle rehabilitation device. Simulation experiments conducted with a virtual prototype in MD ADAMS software validated the performance of the PID-type controller [95]. Azar et al. introduced an algorithm and an improved rule for controlling a lower-limb rehabilitation system. This system was implemented on a 6DoF Stewart parallel robot. Both impedance control and adaptive

control methods were used, and the control parameters were estimated and optimized using artificial neural networks and genetic algorithms. Safety was assured by enabling adaptation under stable conditions. Simulations carried out using MATLAB/SIMULINK demonstrated the effectiveness of this approach when compared to common methods [96].

In 2017, Rosado et al. explored the use of PID controllers for conducting passive rehabilitation exercises. They designed and constructed an ankle rehabilitation prototype that employed a 2-RRSP parallel mechanism. Computer programs for guiding the rehabilitation exercises were developed using open-source software. The prototype facilitated passive exercises that involved path planning and PID control for various ankle movements [97]. Rosado et al. also presented the application of impedance controllers in active rehabilitation exercises. As in the previous study, a parallel ankle rehabilitation prototype using a 2-RRSP mechanism was utilized. They categorized active rehabilitation exercises and user-operation resistance effects into low, medium, and high opposition levels [98]. Du et al. introduced design modeling for a novel 3-RRS spherical parallel mechanism specifically intended for ankle rehabilitation. The study established the kinematics, degree-of-freedom calculations, and inverse kinematics of the mechanism. Multiple inverse solutions were derived, and a forward position analysis method suitable for motor position control was developed. The ankle rehabilitation robot was versatile and applicable in various settings, including homes, hotels, and fitness centers [99]. Zhang et al. highlighted innovative features of a Compliant Ankle Rehabilitation Robot (CARR) that offered an adjustable workspace and torque capacity. The CARR consisted of three rotational degrees of freedom (DoF) and was redundantly actuated by four compliant actuators. Due to the use of a parallel mechanism and compliant actuators, the robot faced the challenge of reconciling conflicting workspace and actuation torque requirements. To address this, the CARR was designed with reconfigurability, allowing it to balance workspace and torque capacity and meet diverse training needs. Theoretical analyses suggested the potential for varying kinematic and dynamic performance by reconfiguring the actuator layout [100].

In 2018, Liao et al. proposed a novel hybrid ankle rehabilitation robot composed of both serial and parallel components. The kinematic performance of this hybrid robot was analyzed, with its parallel part simplified as a constrained 3-PSP mechanism. Mathematical modeling based on screw theory was employed to establish a mathematical model for this component. Inverse kinematics were determined, and factors such as reciprocal twists, Jacobian matrices, and singularities were examined. The study predicted that the workspace of the central point on the moving platform can be expanded while eliminating singularities, making the robot suitable for clinical applications [15]. Rastegarpanah et al. presented a nine-degrees-of-freedom hybrid parallel mechanism designed for ankle rehabilitation, aiming to achieve precise movement in the lower extremities. The methodology for determining stiffness involved calculating the position vectors of each actuator in specific poses using inverse kinematics, thereby obtaining the magnitude and direction of the applied forces. The study leveraged both the stiffness and workspace attributes of parallel robots for ankle rehabilitation. Comparisons were made with standard parallel mechanisms, and the stiffness was evaluated through simulation, which was then compared to a prototype hybrid model in various scenarios [17]. Jamwal et al. investigated the feasibility of a wearable ankle robot for in-home rehabilitation. They began with an analysis of existing technologies and solutions. The complexities of human–robot interactions during rehabilitation were addressed through a fuzzy-logic-based controller designed for ankle treatment. The team proposed visual haptic interfaces to enhance patient engagement and considered web-based communication channels between users and remote physiotherapy staff. The software architecture included patient databases, a graphical user interface, and exercise libraries, thereby ensuring user-friendly operations and offering virtual reality-specific exercises for ankle rehabilitation [101].

In 2019, Wang et al. developed a robot to assist with ankle joint rehabilitation. This robot featured upper and lower platforms connected by a ball–pin pair and driving branches based on an SPS mechanism. Despite having only two degrees of freedom, the upper plat-

form could emulate three types of ankle joint movements. The robot utilized a central ball–pin pair to mimic the natural motions of a patient's foot and ankle joint, while rigid–flexible hybrid driving systems ensured decoupled dorsiflexion/plantar flexion and varus/valgus movements to minimize the risk of secondary injuries [102]. Naruhmi et al. presented a comprehensive analysis of a reconfigurable 3-(rR)PS metamorphic parallel mechanism, exploring its complete workspace and operational modes. The mechanism consisted of three (rR)PS legs, each having an (rR) joint comprising two perpendicular revolute joints. One axis of the (rR) joint could be continuously reconfigured, leading to three distinct configurations. Algebraic geometry techniques were used to derive constraint equations, and primary decomposition computations were carried out for each configuration. The study revealed that the mechanism could exhibit one or two operational modes, depending on the arrangement of the second axes of the (rR) joints. The work characterized the orientation workspaces for both modes and compared the moving platform's instantaneous motion along the same trajectories. An identification approach was also introduced for determining which operational mode corresponded to a given mechanism pose, thereby offering a useful tool for trajectory planning [103].

In 2020, Zuo et al. introduced a wearable parallel mechanism aimed at enhancing the range of equipment options for patients with ankle disabilities. The team conducted kinematic analyses that included inverse position solutions and Jacobian matrices. Performance indices such as reachable workspace, motion isotropy, force transfer capabilities, and maximum torque were developed based on these kinematic solutions. The mechanical structure of the wearable parallel robot was designed to provide motion isotropy, effective force transfer, and high torque capabilities within a broad workspace suitable for ankle rehabilitation [14]. Li et al. concentrated on improving the accuracy and effectiveness of ankle rehabilitation through innovative control strategies. They applied these to a newly developed parallel ankle rehabilitation robot featuring a unique 2-UPS/RRR mechanism, as shown in Figure 8a. The team established the kinematic model for the mechanism and derived both the inverse positional solution and velocity mapping needed for trajectory tracking during passive rehabilitation exercises. Experiments were conducted to determine a torque threshold for detecting the ankle joint's motion intentions, leading to the proposal of an active rehabilitation training strategy. Trials involving healthy subjects showed that the control strategies effectively minimized trajectory-tracking errors during passive exercises and allowed the ankle rehabilitation robot to drive the ankle joint smoothly and flexibly in the intended motion direction. These results confirmed the effectiveness of the proposed control strategies for ankle rehabilitation training [104]. Russo et al. designed the CABLEankle, a lightweight wearable device for ankle motion assistance in both rehabilitation and training, as illustrated in Figure 8b. Employing a cable-driven S-4SPS parallel architecture, the CABLEankle provided motion assistance across a wide range of ankle movements during walking. The design of the mechanism was scrutinized through kinematic and static models, as well as force and workspace analyses. Numerical simulations were conducted to assess the feasibility of the proposed design, specifically characterizing its unique range of ankle motion [105]. Li et al. introduced a new parallel ankle rehabilitation robot with an emphasis on simplicity and nonredundant actuators. They formulated the inverse position solution to calculate the robot's workspace and derived Jacobian matrices for velocity and force mapping. Performance indices such as motion isotropy, force transfer ratio, and force isotropic radius were defined. The inverse dynamic model was developed using the Newton–Euler formulation. The researchers proposed a dynamic evaluation index, called "dynamic uniformity", based on the Jacobian and inertia matrices. A workspace analysis deemed the robot's range suitable for ankle rehabilitation, aligning well with human ankle joint motion. Kinetostatic and dynamic performance analyses confirmed favorable motion properties, high force transfer, and dynamic dexterity, particularly in the central workspace. A simulation validated the rehabilitation process and the inverse dynamic model, underscoring the robot's potential versatility in ankle rehabilitation [106].

In 2021, Pulloquinga et al. discussed the singular configurations of parallel robots (PRs), which can result in control loss. The authors introduced a proximity detection index for type II singularities, aimed at identifying the contributing kinematic chains based on the angle between output twist screws. An experimental benchmark was presented, and PR configurations were analyzed where the proposed index was zero but the forward Jacobian determinant was not. The study illuminated the challenges of handling external actions applied to the PRs in such configurations, highlighting the complexities introduced by manufacturing tolerances [107].

In 2022, Lui et al. introduced a novel ankle rehabilitation robot based on a 2-UPU/RPU parallel mechanism, which they termed the Multi-Locomotion Mode Ankle Rehabilitation Robot (MLMARR). This robot offered various rehabilitation training modes that went beyond basic ankle motion orientation, including up/down and back/forth traction training. The study covered analyses of the degrees of freedom, inverse position solutions, the identification of kinematic singularities, and workspace determination. It also included dynamic simulations aimed at optimizing linear actuators. Experimental data from a prototype MLMARR validated the effectiveness and accuracy of the proposed training modes, showcasing their potential to advance therapies in ankle rehabilitation [108].

In 2023, the field of robotic rehabilitation witnessed significant advancements. Valencia-Segura et al. championed an affordable ankle rehabilitation device, boasting an optimized spherical mechanism with a singular degree of freedom (Figure 8c). This device was tailored for a spectrum of ankle movements, addressing diverse parameters such as motion angle and force transmission [109]. On a parallel note, Wu et al. showcased a groundbreaking 9DoF robotic system aimed at patients with balance disorders. This robot, integrating a series-parallel hybrid motion platform, was constructed from two symmetrically arranged sets of motion platforms, each containing a 6DoF vestibular parallel device and a 3DoF proprioception parallel device. This unique configuration allowed for DoF decoupling and swift responses, leading to a structure optimal for vestibular and proprioceptive simulation. with modularized vestibular and proprioceptive components, the robot's flexibility and capabilities were significantly amplified [110]. Concurrently, Shi et al. enriched the sector with their 5DoF serial–parallel hybrid robot designed for gait and balance training. This robot integrated a 2DoF gait generator employing the H-Bot mechanism and a 3DoF parallel balance module, offering natural balance exercises by aligning the footplate's rotation center with the ankle joint. with embedded sensors, the robot achieved precise closed-loop control for both position and orientation. Post-construction tests demonstrated its prowess in accurate tracking, solidifying its potential as a holistic platform for combined gait and balance rehabilitation. Collectively, these studies illuminate the rapid progress in the domain of robotic rehabilitation, paving the way for sophisticated and patient-centric solutions [111]. Zuo et al. (2023) introduced a groundbreaking design for composite patient-external fixators (PEFs), emphasizing precise correction requirements. By employing multi-site frame mounting and a diverse set of struts, they demonstrated the feasibility of their method, backed by clinical cases and simulations, ensuring that the device could adapt to various deformities and maintain stability [112].

In parallel, Zermane et al. (2023) presented an innovative 3-PRS robotic platform tailored for human balance assessment and rehabilitation. The platform's design, which encompassed rotational and vertical movements, underwent rigorous optimization using multiple algorithms, ultimately pinpointing the optimal geometric parameters [113]. Lastly, Zhang et al. (2023) ventured into the realm of ankle–foot motion, proposing a new evaluation index called the "ankle-foot motion comfort zone". By integrating this index into their optimization method for a 3DoF ankle rehabilitation robot, they not only underscored the interconnection of ankle–foot motion but also showcased the adaptability and efficacy of their method. Collectively, these studies indicate a promising trajectory in the development and optimization of rehabilitation devices, potentially revolutionizing patient care in orthopedics and balance disorders [114].

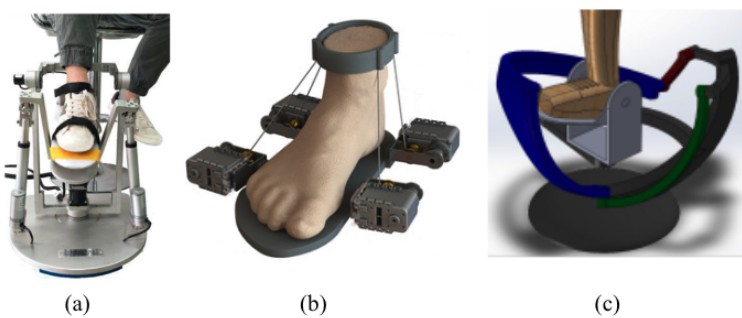

(a)             (b)             (c)

**Figure 8.** Ankle joint mechanisms: (**a**) 2-UPS/RRR mechanism by Beijing University [104]; (**b**) S-4SPS mechanism by the University of Nottingham [105]; (**c**) spherical mechanism by Instituto Politécnico Nacional, Mexico [109]. All are available under a Creative Commons Attribution License.

**Table 6.** Parallel robots for the rehabilitation of the ankle joint.

| Author | Year | Country | Device | TRL | Mechanism | DoF | ToM | Actuator | Model | Tool |
|---|---|---|---|---|---|---|---|---|---|---|
| Wang et al. [88] | 2012 | China | AR | 2 | 3-SPS/SP | 3 | PD, IE, AA | NS | FIK | M |
| Saglia et al. [89] | 2013 | Italy | AR | 4 | 3-UPS/U | 2 | PD, IE | EL | FK | NS |
| Wang et al. [90] | 2013 | China | AR | 2 | 3-RUS/RRR | 3 | PD, IE, AA | ER | IK | AD |
| Jamwal et al. [91] | 2014 | India | AR | 4 | PMA | 3 | PD, IE, AA | PL | FK | M |
| Jamwal et al. [92] | 2015 | India | AR | 2 | PMA | 3 | NS | PM | FIK | M |
| Vallés et al. [93] | 2015 | Spain | AR | 3 | 3-PRS | 3 | PD, IE | EL | IK | ROS |
| Jamwal et al. [94] | 2016 | Kazakhstan | AR | 4 | PMA | 3 | PD, IE, AA | PM | FK | NS |
| Azar et al. [96] | 2016 | Iran | AR | 3 | 6-UPS | 6 | NS | PM | IK | NS |
| Ruiz-Hidalgo et al. [95] | 2016 | Mexico | AR | 3 | 3-SPR | 3 | NS | EL | IK | AD |
| Rosado et al. [97] | 2017 | Mexico | AR | 2 | 2-RRSP | 2 | PD, IE | ER | NS | M |
| Rosado et al. [98] | 2017 | Mexico | AR | 3 | 2-RRSP | 2 | PD, IE | ER | NS | AD |
| Du et al. [99] | 2017 | China | AR | 2 | 3-RRS | 3 | PD, IE | EL | IK | NS |
| Zhang et al. [100] | 2017 | New Zealand | AR | 2 | 4-PMA | 3 | PD, IE, AA | PM | IK | NS |
| Liao et al. [15] | 2018 | China | AR | 2 | 3-PSP | 3 | PD, IE | NS | IK | NS |
| Rastegarpanah et al. [17] | 2018 | UK | AR | 3 | 6-UPS/3SPR | 9 | NS | NS | IK | NS |
| Jamwal et al. [101] | 2018 | Kazakhstan | AR | 4 | PMA | 3 | PD, IE, AA | PM | IK | LV |
| Wang et al. [102] | 2019 | China | AR | 3 | 2-SPS | 2 | PD, IE | EL | FK | NS |
| Naruhmi et al. [103] | 2019 | Indonesia | AR | 3 | 3(rR)PS | 3 | NS | NS | FK | NS |
| Zuo et al. [14] | 2020 | China | AR | 2 | 2-UPS/RRR | 3 | PD, IE, AA | EL | IK | NS |
| Li et al. [104] | 2020 | China | AR | 3 | 2-UPS/RRR | 3 | PD, IE, AA | EL | IK | NS |
| Russo et al. [105] | 2020 | UK | AR | 2 | cable-driven | 3 | PD, IE, AA | EL | NS | NS |
| Li et al. [106] | 2020 | China | AR | 3 | 2-UPS/RRR | 3 | PD, IE, AA | EL | IK | M |
| Pulloquinga et al. [107] | 2021 | Spain | AR | 3 | 3-UPS/RPU | 4 | NS | EL | IK | M, LV |
| Liu et al. [108] | 2022 | China | AR | 2 | 2-UPU/RPU | 2 | PD, IE | PM | IK | NS |
| Valencia-Segura et al. [109] | 2023 | Mexico | AR | 3 | 4-bar | 2 | PD, IE, AA | NS | IK | CAD |
| Wu et al. [110] | 2023 | China | AR | 4 | 6-SSP, 3-RPS | 9 | PD, IE, AA | EL | IK | NS |
| Shi et al. [111] | 2023 | China | AR | 4 | 3-RRR | 3 | PD, IE, AA | ER | IK | M |
| Zuo et al. [112] | 2023 | China | AR | 4 | RPUR, SPR | 5 | PD, IE, AA | ER | IK | NS |
| Zermane et al. [113] | 2023 | France | AR | 4 | 3-PRS | 3 | PD, IE, AA | ER | IK | SA |
| Zhang et al. [114] | 2023 | China | AR | 4 | 2-UPU/PRPS | 3 | PD, IE, AA | EL | IK | M |

Abbreviations: AR—ankle rehabilitation, 3-SPS/SP—3 (spherical-prismatic-spherical)/ 1(spherical-prismatic), 3-UPS/U—3 (universal-prismatic-spherical)/1 (universal), 3-RUS/RRR—3 (revolute-universal-spherical)/1 (revolute-revolute-revolute), PMA—pneumatic muscle actuator, 3-SPR—3 (spherical-prismatic-revolute), 3-PRS—3 (prismatic-revolute-spherical), 2-RRSP—2 (revolute-revolute-spherical-prismatic), 3-RRS—3 (revolute-revolute-spherical), 6-UPS—6 (universal-prismatic-spherical), 4-PMA—4 (pneumatic muscle actuator), 3-PSP—3 (prismatic-spherical-prismatic), 6-UPS/3SPR—6 (universal-prismatic-spherical)/3 (spherical-prismatic-revolute), 2-SPS—2 (spherical-prismatic-spherical), 2-UPS/RRR—2 (universal-prismatic-spherical)/(revolute-revolute-revolute), 2-UPS/RRR—2 (universal-prismatic-spherical)/1 (revolute-revolute-revolute), 3(rR)PS—two perpendicular revolute joints, 3-UPS/RPU—3 (universal-prismatic-spherical)/(revolute-prismatic-universal), 2-UPU/RPU—2 (universal-prismatic-universal)/1 (revolute-prismatic-universal), 3RRR—3 ( revolute-revolute-revolute), RPUR—(revolute- prismatic-universal-revolute), SPR—(spherical- prismatic-revolute), 3-PRS—3 (prismatic-revolute-spherical), 2-UPU/PRPS—2 (universal-prismatic-universal)/(prismatic-revolute-prismatic-spherical), PD—plantarflexion–dorsiflexion, IE—inversion–eversion, AA—abduction–adduction, NS—not specified, ER—electric rotary, EL—electric linear, PL—pneumatic linear, PM—pneumatic muscle, IK—inverse kinematic, FK—forward kinematic, FIK—forward and inverse kinematic, M—MATLAB, AD—Adams, LV—LabVIEW, SA—simulated annealing, ROS—Robot Operating System, CAD—Computer-Aided Design.

## 5. Discussion

### 5.1. Timeline of Parallel Robots from 2012 to 2023

This comprehensive timeline (Figure 9) reflects the rapid advancements in robotics tailored to medical and humanoid applications over a decade. The expansive body of research in robotic technology showcases advancements across rehabilitation, assistance, and humanoid applications.

- Rehabilitation focus: Rehabilitative robotics present a variety of versatile solutions, from ankle robots equipped with impedance control and redundant actuation, to teleoperable, modular ankle devices and multi-objective-optimized wearable ankle robots. Shoulder and wrist rehabilitation benefit from adaptive controllers, multi-DoF prosthetics, and anti-vibration control technologies.
  Ankle rehabilitation has been a consistent area of interest every year, with advances in control algorithms [89], wearable designs [91–93], hybrid mechanisms [15], natural motion mimicry [102], and high-torque applications [14,104–106]. As of 2023, rehabilitation techniques have expanded to treat balance disorders [110] and offer solutions for vertigo diagnosis [113].
- Cable-driven and parallel robots: Starting in 2012 with applications like shoulder movement simulation and ankle rehabilitation, there has been a clear focus on cable-driven and parallel mechanisms. These designs emphasize human mimicry, offering smooth and organic movements. Examples include neck movements [39–41], wrist motion [68,74], and shoulder movements [45,53]
  In terms of mimicking human biomechanics, cable-driven robots excel in recreating neck and cervical spine movements, while various exoskeletons and prosthetics, such as 7DoF shoulder and arm mechanisms, offer enhanced spatial characteristics.
- Humanoid design: The timeline indicates the progression in bionic and humanoid designs. From bionic joints in 2013 [54] to more comprehensive hybrid humanoid arms in 2021 [59,59], there has been an evident push towards creating robots that closely emulate human characteristics. The introduction of LARMbot 2 in 2018 [87] demonstrates the practical implementations of these humanoid designs.
  Humanoid robots feature innovations like bionic joints, pneumatic-muscle-driven manipulators, and multi-objective-optimized trajectory controls, designed for speed, stability, and energy efficiency.
- Neck and shoulder rehabilitation: Starting from 2013 with the cable-driven robot mimicking human neck movements [39], there has been a sustained focus on neck and shoulder rehabilitation mechanisms over the years. By 2021, there was a move towards more flexible and soft mechanisms for neck rehabilitation [3], further improving in 2023 with cable-driven exoskeletons for cervical rehabilitation [35].
- Wrist and hand innovations: Over the timeline, wrist innovations have ranged from modular designs [68] to stiffness-optimizable exoskeletons [49] and cable-driven prostheses [72]. The emphasis on wrist rehab and prostheses highlights the importance of restoring hand functions in patients.
- Hybrid designs: The later years (especially post-2020) showcase a strong interest in hybrid systems. For instance, in 2023, highlights include series-parallel hybrid modular joint humanoids [60] and serial–parallel hybrid robots [111].
- Specialized solutions: Some innovations are particularly noteworthy because they address specific medical conditions or challenges, like the EMG-recording dynamic neck brace for amyotrophic lateral sclerosis (ALS) [2] and multi-segment foot and ankle deformity correction in 2023 [112].
- Advanced control and optimization techniques: The use of genetic algorithms [45, 46,51], neural networks [22,82,96], and Koopman controllers [64,67] indicates the integration of sophisticated computational techniques in optimizing and controlling robotic movements.

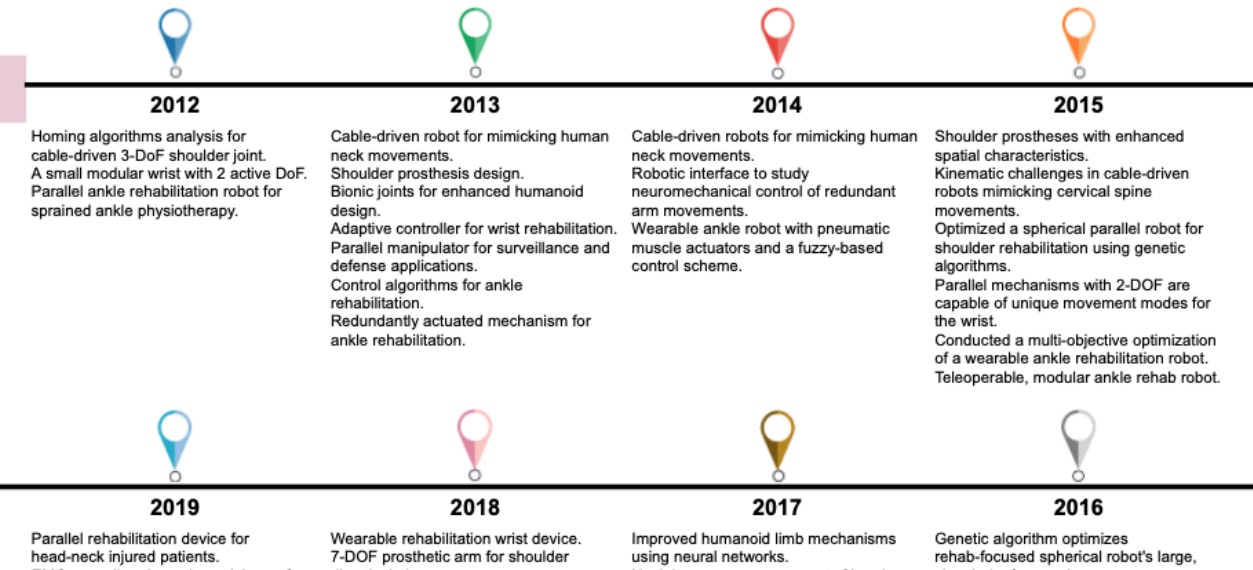

**Figure 9.** Timeline of parallel robots from 2012 to 2023. References: 2012 [53,68,88], 2013 [39,50,54, 61,73,89,90], 2014 [40,41,44,91], 2015 [20,42,45,74,92,93], 2016 [24,46,47,55,80,94–96], 2017 [22,36,43, 56,62,75,86,97–100], 2018 [6,15,17,19,31,37,51,52,69,76,87,101], 2019 [1,2,21,27,38,57,58,77,83,102,103], 2020 [9,10,14,33,48,81,84,104–106], 2021 [3,11,16,18,26,49,59,63,70,107], 2022 [4,64,71,72,78,79,82,108], 2023 [35,60,65–67,85,109–114].

## 5.2. Advancements in Parallel Robots from 2012 to 2023

This section presents an analysis of articles published on parallel robots in the fields of rehabilitation, assistance, and humanoid robotics between the years 2012 and 2023, across various countries. Tables 7–9 provide information on the distribution of articles based on the targeted body parts (e.g., neck, shoulder, wrist, hip, and ankle). The "Total" column indicates the sum of articles related to each country, while the "Percentage" column shows the proportion of articles contributed by each country.

- Parallel robots for rehabilitation:

  Firstly, it is evident that China has emerged as a significant contributor to research in this field, accounting for 37.3% of the total articles. This underscores China's robust presence in parallel robot research, particularly in the area of rehabilitation for various joints.

  Mexico accounts for 9.8% of the total articles. This underscores the active participation of researchers from this country in investigating the applications of parallel robots, especially in the neck and ankle areas. On the other hand, each of the following countries: the United States, the United Kingdom (UK), and Australia, accounts for 7.8% of the total articles.

  Moreover, while individual countries like India, Iran, each contributed a smaller percentage, they collectively made up a significant portion of the total articles, at 5.9% each. This fact demonstrates a global interest and investment in advancing the field of parallel robot rehabilitation.

  Conversely, some countries, such as Spain and Kazakhstan, accounting for 3.9%, have demonstrated a more balanced distribution of articles across different joint areas, indicating a diverse research focus.

  It is important to note that there are countries, such as Japan, Italy, New Zealand, Indonesia and France, that each contributed to the total with a smaller percentage, starting from 2.0%. While their contributions may be comparatively modest, they nonetheless play a role in enriching the overall body of knowledge in the field of parallel robot applications for rehabilitation.

**Table 7.** Analysis of articles published on parallel robots for rehabilitation from 2012 to 2023 across various countries.

| Country | Neck | Shoulder | Wrist | Hip | Ankle | Total | Percentage |
|---------|------|----------|-------|-----|-------|-------|------------|
| India | 1 | - | - | - | 2 | 3 | 5.9% |
| USA | 1 | 2 | 1 | - | - | 4 | 7.8% |
| Mexico | 1 | - | - | - | 4 | 5 | 9.8% |
| UK | - | 1 | - | 1 | 2 | 4 | 7.8% |
| Iran | - | 2 | - | - | 1 | 3 | 5.9% |
| Australia | - | 1 | 3 | - | - | 5 | 7.8% |
| China | 1 | - | 2 | 3 | 13 | 18 | 37.3% |
| Japan | - | - | 1 | - | - | 1 | 2.0% |
| Italy | - | - | - | - | 1 | 1 | 2.0% |
| Spain | - | - | - | - | 2 | 2 | 3.9% |
| Kazakhstan | - | - | - | - | 2 | 2 | 3.9% |
| New Zealand | - | - | 1 | - | - | 1 | 2.0% |
| Indonesia | - | - | 1 | - | - | 1 | 2.0% |
| France | - | - | 1 | - | - | 1 | 2.0% |

- Parallel robots for assistance:

  The United States (USA) leads in contributions to research in this field, accounting for 31.3% of the total articles. This statistic suggests that the USA is vigorously engaged in research on parallel robots for assistance, with particular emphasis on the neck, shoulder, and wrist areas. China follows closely, contributing 25.0% of the articles, with a focus on assistance for the neck and hip.

  Japan also has a significant presence, contributing 12.5% of the articles and concentrating primarily on shoulder assistance. This highlights Japan's active involvement in research concerning parallel robots designed for assistance purposes.

  The analysis revealed that multiple countries, including Italy, Mexico, South Korea, and the United Kingdom (UK), each contributed 6.3% to the total body of articles. These articles were predominantly centered on shoulder and wrist assistance.

  The data indicate a widespread interest in parallel robots for assistance on a global scale. While the USA, China, and Japan emerged as the major contributors, other

countries are also actively participating in research in this field, showing a dispersed research effort.

The lack of contributions focusing on certain body parts, such as the ankle, in some countries may suggest that the research in those nations is primarily oriented toward other specific areas of assistance.

**Table 8.** Analysis of articles published on parallel robots for assistance from 2012 to 2023 across various countries.

| Country | Neck | Shoulder | Wrist | Hip | Ankle | Total | Percentage |
|---|---|---|---|---|---|---|---|
| USA | 2 | 1 | 2 | - | - | 5 | 31.3% |
| China | 1 | - | - | 3 | - | 4 | 25.0% |
| Japan | - | 2 | - | - | - | 2 | 12.5% |
| Italy | - | 1 | - | - | - | 1 | 6.3% |
| Mexico | - | 1 | - | - | - | 1 | 6.3% |
| South Korea | - | - | 1 | - | - | 1 | 6.3% |
| Korea | - | - | 1 | - | - | 1 | 6.3% |
| UK | - | - | 1 | - | - | 1 | 6.3% |

- Parallel robots for humanoids:

  China is the most prominent contributor to research in this field, responsible for 76.9% of the total articles published. This high percentage underscores a significant focus on parallel robots for humanoids within the Chinese research community. In terms of specific body parts, China's contributions span articles related to the neck, shoulder, wrist, and hip.

  Several other countries, including France, Denmark, Mexico, the United Kingdom (UK), Turkey, and Italy, have also made contributions, albeit to a lesser extent. Each of these countries accounts for 3.8% of the total number of articles.

**Table 9.** Analysis of articles published on parallel robots for humanoids from 2012 to 2023 across various countries.

| Country | Neck | Shoulder | Wrist | Hip | Ankle | Total | Percentage |
|---|---|---|---|---|---|---|---|
| China | 6 | 6 | 6 | 2 | - | 20 | 76.9% |
| France | - | 1 | - | - | - | 1 | 3.8% |
| Denmark | - | 1 | - | - | - | 1 | 3.8% |
| Mexico | - | - | 1 | - | - | 1 | 3.8% |
| UK | - | - | 1 | - | - | 1 | 3.8% |
| Turkey | - | - | 1 | - | - | 1 | 3.8% |
| Italy | - | - | - | 1 | - | 1 | 3.8% |

*5.3. Technology Readiness Levels—TRLs*

Technology readiness levels (TRLs) are used to gauge the maturity level of a technology as it progresses through its research, development, and deployment phases. TRLs are quantified on a scale from 1 to 9, with level 9 representing the most mature technology [115].

Tables 10–12 present an analysis of the technology readiness levels (TRLs) for parallel robots designed for rehabilitation, assistance, and humanoid applications. These tables serve as indicators of a given technology's maturity, spanning from early-stage conceptual models (TRL 1) to fully operational systems (TRL 9). Each table details the distribution of TRLs across various body parts, including the neck, shoulder, wrist, hip, and ankle. The "Total" column lists the cumulative count of articles for each TRL level, while the "Percentage" column indicates the proportion of articles that correspond to each specific TRL.

- Parallel robots for rehabilitation:

  The lack of articles at TRL 1 indicates that most research publications have moved beyond foundational scientific principles to more advanced stages. with 25.49% of the

total articles, TRL 2 serves as a transitional phase from theoretical concepts to practical experimentation. The notable concentration at TRL 2 suggests that researchers are keenly exploring and prototyping new approaches to rehabilitation using parallel robots, especially for the shoulder and ankle regions.

At 45.10%, TRL 3 boasts the largest percentage of articles, indicating that a significant volume of research has progressed to the prototype-building and proof-of-concept stages. The range of body parts covered (neck, shoulder, wrist, hip, and ankle) demonstrates a broad investigation into the applications of parallel robots in various rehabilitation contexts.

Articles at TRL 4 make up 29.41% of the total publications. These studies imply that some research endeavors have reached the level of system integration and functional testing. The existence of articles at this TRL suggests the ongoing refinement of parallel robot systems, potentially bringing them closer to practical implementation in rehabilitation settings.

**Table 10.** Technology readiness levels of parallel robots for rehabilitation.

| TRL | Neck | Shoulder | Wrist | Hip | Ankle | Total | Percentage |
|-----|------|----------|-------|-----|-------|-------|------------|
| TRL 1 | - | - | - | - | - | 0 | 0% |
| TRL 2 | - | 3 | - | - | 10 | 13 | 25.49% |
| TRL 3 | 2 | 2 | 6 | 2 | 11 | 23 | 45.10% |
| TRL 4 | 2 | 1 | 2 | 1 | 9 | 15 | 29.41% |

- Parallel robots for assistance:

  TRL 1 has no corresponding articles, suggesting that publications are primarily focused on concepts beyond initial scientific principles. Articles at TRL 2 account for 25.0% of the total, indicating a transition in research efforts toward practical experimentation and validation, particularly in the contexts of shoulder and wrist assistance.

  With 68.75% of the total articles classified under TRL 3, the research community has made significant strides in developing prototypes and demonstrating the feasibility of parallel robot systems for assistance across various body parts. The high concentration of articles at TRL 3 signifies a strong emphasis on constructing functional prototypes with the potential for real-world applications.

  Of the total articles, 6.25% are classified under TRL 4. This may indicate that research has advanced to the stages of system integration, advanced testing, or practical implementation.

  The distribution of TRLs across different body parts provides insights into the areas that have garnered more attention in terms of technological development for assistance. While neck, shoulder, and wrist assistance have seen notable progress, ankle assistance has not yet advanced beyond TRL 2.

**Table 11.** Technology readiness levels of parallel robots for assistance.

| TRL | Neck | Shoulder | Wrist | Hip | Ankle | Total | Percentage |
|-----|------|----------|-------|-----|-------|-------|------------|
| TRL 1 | - | - | - | - | - | 0 | 0% |
| TRL 2 | - | 2 | 1 | 1 | - | 4 | 25.0% |
| TRL 3 | 3 | 3 | 4 | 1 | - | 11 | 68.75% |
| TRL 4 | - | - | - | 1 | - | 1 | 6.25% |

- Parallel robots for humanoids:

  Articles at TRL 1 and TRL 2 together account for a collective 50.0% of the total. This suggests a substantial focus on both theoretical groundwork (TRL 1) and practical experimentation (TRL 2). Both neck and shoulder areas have received particular attention at these early stages.

Articles classified under TRL 3 constitute the largest percentage, making up 46.15% of the total. This indicates that researchers have moved beyond initial theoretical concepts to the active development and validation of prototypes. The distribution across various body parts reflects a comprehensive focus on the application of parallel robots in humanoid systems.

There are 3.85% corresponding articles at TRL 4, suggesting that research has advanced to the stage of system integration or operational testing.

The distribution of TRLs across different body parts provides insights into the areas that have received the most attention in terms of technological development for humanoid applications. Notable progress has been observed across TRLs for the shoulder and wrist.

**Table 12.** Technology readiness levels of parallel robots in the area of humanoids.

| TRL | Neck | Shoulder | Wrist | Hip | Ankle | Total | Percentage |
|-----|------|----------|-------|-----|-------|-------|------------|
| TRL 1 | 2 | - | - | - | - | 2 | 7.69% |
| TRL 2 | 3 | 3 | 4 | 1 | - | 11 | 42.31% |
| TRL 3 | 1 | 4 | 5 | 2 | - | 12 | 46.15% |
| TRL 4 | - | 1 | - | - | - | 1 | 3.85% |

*5.4. Design, Number of Degrees of Freedom, and Kinematics Structure in Parallel Robotics*

The intricacies of designing and optimizing parallel robotic systems constitute an area of intense research and development. This arena is particularly significant because the ultimate performance of a robot—be it in rehabilitation, assistance, or humanoid constructs—is deeply rooted in its initial design parameters and subsequent optimization techniques.

### 5.4.1. Design Principles

The design of parallel robots for use in rehabilitation, assistance, and humanoid applications must account for the specific requirements of each application. Each joint of the human body—such as the neck, shoulder, wrist, hip, and ankle—has unique motion characteristics and specific demands in terms of degrees of freedom, range of motion, required force, and precision [45]. It is, therefore, necessary to tailor the robot's design to each specific joint while considering the patient's needs and rehabilitation goals.

The design process includes structural analysis as well as the selection of actuators and sensors [62]. This allows for the creation of customized designs and a wide variety of configurations, enabling the robots to perform movements like translations, rotations [90], inclinations, or a combination thereof.

Compared to serial robots, parallel robots are designed to be more precise and rigid. This is attributable to the multiple kinematic chains acting in parallel, which offer greater stability [76] and resistance to deformation [98].

Analytically, parallel mechanisms pose additional challenges due to their inherent complexity. The kinematic and dynamic analysis of these mechanisms requires solving nonlinear equations [49,56,91] and constraints and considering interactions between the kinematic chains. Such mechanisms also face limitations in design and control complexity, the need for appropriate actuators and precise sensors [62], associated costs, and technology readiness levels.

### 5.4.2. Number of Degrees of Freedom and Kinematic Structure

The relationship between the degrees of freedom (DoF) and kinematic structures is foundational in the field of parallel robotics and mechanical design. In essence, the degrees of freedom of a system define the number of independent parameters that determine its configuration. Kinematic structures, on the other hand, describe the arrangement and connectivity of joints and links in a mechanical system. The nature of this arrangement—how parts are interconnected—determines the system's degrees of freedom. For instance, Table 13 presents a comprehensive overview of the degrees of freedom (DoF) and the

corresponding kinematic structures associated with various human joints, with a particular emphasis on their utilization in rehabilitation, assistance, and humanoid robot technologies.

- Neck joint:

  For the neck joint, the dominant kinematic structures are primarily based on a 3DoF framework. In the rehabilitation sphere, there are models such as 3-RPS [1] and 3-RRR [2]. In the context of assistance robots, configurations like 3-RRS [36], 3-RRR [37], and 3-RXS [38] are used. Humanoid robotics often favor cable-driven mechanisms [3,39,41–43]. A 4DoF structure, represented by the 4-SPS model [4], is also common in the rehabilitation sector.

- Shoulder joint:

  Given its complex nature, the shoulder joint exhibits a diverse range of kinematic structures. In humanoid robots, 2DoF mechanisms like 2-UPUR/RU [55] and cable-driven [56] systems are prominent. As the DoF expand to 3 and 5, the rehabilitation domain presents an array of models, including 5R [44], 3-RSS/S [45], and SPM [47]. In the assistance robotics realm, structures such as 3-SPS/P [20], 3-RSS/S [46], and 3-RRR [19,51] emerge. The 7DoF mechanisms, including designs like 4B-SPM [49] and cable-driven [50] methods, underscore the sophistication of this joint's modeling.

- Wrist joint:

  Given its essential function in facilitating dexterity, the wrist showcases a broad spectrum of kinematic structures. In rehabilitation for a 2DoF wrist, the 3-RPS [61] configuration stands out. In humanoid applications, models evolve in complexity from 2-PUU/RPS [76] for 2DoF to designs like 5-SPM [75] for 5DoF. The ubiquitous presence of cable-driven [72] solutions across varying DoF emphasizes their role in ensuring flexibility and adaptability.

- Hip joint:

  For the 2DoF hip joint, while the rehabilitation and assistance domains remain relatively unexplored, humanoid designs have proposed models like 4-SPS/CU [86]. A 3DoF hip favors models like 2-UPS/RRR [81,82] in rehabilitation and 4-RPS [84] in assistance robotics. Interestingly, the 6DoF hip primarily features the 6-UPS [80,83] configuration, evident in both the rehabilitation and assistance domains.

- Ankle joint:

  The ankle, crucial for locomotion, has garnered significant attention in the rehabilitation sector. The multitude of designs for a 3DoF ankle, ranging from 3-SPS/SP [88] to 3-PSP [15], attests to this. The frequent inclusion of configurations like PMA in multiple references underscores its significance. Higher-DoF designs, such as the 6-UPS [96] for 6DoF and 6-UPS/3SPR [17] for 9DoF, suggest an emphasis on advancing ankle dexterity in robotics.

  The choice of kinematic structure affects other design parameters like the system's strength, flexibility, control complexity, and efficiency. It becomes evident that the technological advances in rehabilitation, assistance, and humanoid robotics are significantly intertwined with the complexities of human anatomy. The extensive research cited above underscores the relentless endeavors of scientists and engineers to emulate human-like movement and function. As robotics continues its foray into human-centric applications, such insights will become pivotal. They not only reflect the current state of art but also hint at the vast potential awaiting in the future of biomechanics and robotics.

**Table 13.** Number of degrees of freedom and kinematic structure.

| Joint | DoF | Rehabilitation | Assistance | Humanoids |
|---|---|---|---|---|
| Neck | 3 | 3-RPS [1], 3-RRR [2], cable-driven [35] | 3-RRS [36], 3-RRR [37], 3-RXS [38] | cable-driven [3,39,41–43] |
| | 4 | 4-SPS [4] | - | - |
| Shoulder | 2 | - | - | 2-UPUR/RU [55], cable-driven [56] |
| | 3 | 5R [44], 3-RSS/S [45], SPM [47], cable-driven [48] | 3-SPS/P [20], 3-RSS/S [46], 3-RRR [19,51], 4B-SPM [52] | 3-RRR [58], 5R [59], cable-driven [35] |
| | 5 | - | - | 5-PMA [54,57] |
| | 7 | 4B-SPM [49] | cable-driven [50] | cable-driven [53] |
| Wrist | 2 | 3-RPS [61] | 2-UPS/S [68], 2-CSPM [70], cable-driven [72] | 3-4R [74], 2-PUU/RPS [76], cable-driven [9], 3-UPS/S [59], 3-RSR/UUP [78], 3-RPS/US [79] |
| | 3 | 2-URR/RRS [62], 3-RRR [6], 6-SPS/PS [63], 4-BMA [66,67], cable-driven [65] | 2-RRR [69], 2-RRRPR/RRPR [71] | 3-SPS/S [73] |
| | 4 | - | - | cable-driven [77] |
| | 5 | - | - | 5-SPM [75] |
| Hip | 2 | - | 2-UPS+S [85] | 4-SPS/CU [86] |
| | 3 | 2-UPS/RRR [81,82] | 4-RPS [84] | |
| | 4 | - | - | 4-SPS [87] |
| | 6 | 6-UPS [80] | 6-UPS [83] | - |
| Ankle | 2 | 3-UPS/U [89], 2-RRSP [97,98], 2-SPS [102], 2-UPU/RPU [108], 4-bar [109] | - | - |
| | 3 | 3-SPS/SP [88], 3-RUS/RRR [90], PMA [91,92,94,101], 3-PRS [93], 3-SPR [95], 3-RRS [99], 4-PMA [100], 3-PSP [15], 3(rR)PS [103], 2-UPS/RRR [14], 2-UPS/RRR [104], cable-driven [105], 2-UPS/RRR [106], 3-RRR [111], 3-PRS [113], 2-UPU/PRPS [114] | - | - |
| | 4 | 3-UPS/RPU [107] | - | - |
| | 5 | RPUR, SPR [112] | - | - |
| | 6 | 6-UPS [96] | - | - |
| | 9 | 6-UPS/3SPR [17], 6-SSP, 3-RPS [110] | - | - |

### 5.5. Assessment of Workspace, Functional Capabilities, and Performance Methods in Parallel Robotics

The systematic evaluation of workspace dimensions, functional capabilities, and performance indices is critical for the continued advancement of parallel robotics. This appraisal serves to deepen our understanding of the complex operational characteristics of these robots, thereby laying the foundation for future technological innovations and tailored application scenarios.

- Workspace:

  Workspace analysis and optimization are recurring themes in a range of robotic technologies, often serving medical and rehabilitation purposes. Various studies have emphasized different methods and objectives to optimize workspace. For instance, Gao et al. [39,41] and Enferadi et al. [45] used simulations and genetic algorithms, respectively, to assess workspace under positive cable tension constraints and maximize it, with Enferadi's work noting the advantage of a singularity-free workspace. Niyetkaliyev's hybrid mechanism [48] covered a full range of shoulder movements in a singularity-free workspace, while Sekine et al. [20] and Song et al. [84] focused on prosthetic arms and hips that offered increased and rotating workspaces, respectively. Zhang et al. [100] employed geometric parameter optimization to prevent workspace

singularities in an ankle rehabilitation robot. The studies of Lee et al. [69] and Serracin et al. [68] considered enhancing workspace efficiency and user comfort, and Wu and Chaparro-Altamirano [73] addressed orientation workspace and workspace calculations. Additional research by Zhang et al. [81], Liao et al. [15], Rastegarpanah et al. [17], Naruhmi et al. [103], and Zuo et al. [14] delved into adjustable workspaces, the expansion of central points on moving platforms, and broad workspaces for ankle rehabilitation, often validated through simulations, prototypes, or theoretical analyses. Overall, the workspace is considered a critical design factor, evaluated and optimized through various methods to meet different performance requirements.

- Functional capabilities:
  The functional capabilities of parallel robots, such as dexterity, manipulability, and range of motion, are critical for their application in rehabilitation and assistive technologies.
  Dexterity: This is often measured using indices like the "manipulability measure", which is derived from the robot's Jacobian matrix [51,72,90,106].
  Manipulability: This represents the ability of the robot to move in different directions [83,90].
  Range of motion: This typically involves analyzing how well the robot can perform tasks that mimic human joint motions, such as rotations and translations [4,6,36,37,70,84,88].

- Performance evaluation methods:
  The performance evaluation of parallel robots is multi-faceted, typically incorporating metrics such as speed [28,59], accuracy [33,53,57,84,104,108], repeatability, load-carrying capacity [79,80], energy efficiency, and safety [96]. Methods for performance evaluation include computational methods, and simulations for statistical analysis and finite element analysis (FEA) for mechanical property evaluation are popular choices [38].
  Furthermore, various software tools were prevalent for simulation and analysis in the studies reviewed. MATLAB, SimScape Multibody [54,78], ADAMS [19,53,59,59,90,95,98], and ANSYS [38] each offer unique capabilities for evaluating different aspects of robot performance, workspace, and function. Their choice often depends on the specific requirements of the study.
  Based on existing research, MATLAB is commonly used to simulate parallel robots because of its matrix-handling abilities, algorithm implementation, function representation, user interface creation, and interoperability. SimScape Multibody, part of MATLAB's Simulink suite, offers block-diagram-based modeling for rigid-body mechanics, describing kinematics and dynamics through blocks representing bodies and joints. This enables quick validation and offers 3D visualization for system movement comprehension. MSC ADAMS facilitates the study of dynamics, load distribution, and forces in mechanical systems by allowing the creation of virtual prototypes underpinned by real-world physics. Meanwhile, ANSYS specializes in kinematic simulations, providing detailed analyses of motion and structural parameters, including rotation angles and force requirements.

*5.6. Material Selection in the Development of Parallel Robotics*

The choice of materials in the fabrication of parallel robotic systems plays a role in meeting targeted performance metrics, durability, and functional objectives. Given the expanding applications of parallel robotics in medical rehabilitation, assistive technologies, and humanoid robots, the criteria for material selection have become increasingly stringent.

- Metals
  Aluminum: This metal is highly prized for its advantageous strength-to-weight ratio, thermal and electrical conductivity, and intrinsic resistance to corrosion. It is often the material of choice for constructing lightweight robotic frameworks. This material was used in a hybrid parallel robot designed for foot rehabilitation [17] and a hybrid exoskeleton for elbow–forearm–wrist rehabilitation [62].

Stainless steel: Renowned for its exceptional durability and corrosion resistance, stainless steel finds utility in applications that demand high mechanical strength and rigidity. This material was utilized in a distal arm exoskeleton for stroke and spinal cord injury rehabilitation [7] and a rope-driven mechanism for wrist rehabilitation [9].

Carbon steel: Medium-carbon steels like 1045 are used in applications where a balance of strength and ductility is required. This material was utilized in a novel bio-inspired and cable-driven hybrid parallel shoulder mechanism [48].

- Plastics

These polymers are commonly selected for components that necessitate low mechanical strength but prioritize lightweight characteristics. ABS (acrylonitrile butadiene tyrene) is a type of thermoplastic polymer. It is a copolymer made from three different monomers: acrylonitrile, butadiene, and styrene. Acrylonitrile provides chemical resistance and thermal stability. Butadiene contributes toughness and impact resistance. Styrene offers rigidity and a glossy finish. This material was employed in a parallel robot designed for a prosthetic arm aimed at shoulder disarticulation [19], a rigid–flexible parallel mechanism for a neck brace [38], a prosthetic arm [51], and a humanoid robot with parallel architectures [87].

- Composite materials

Carbon-fiber composites: Valued for their extraordinary strength-to-weight ratios and rigidity, these composites are ideal for crafting structural components that must be both resilient and lightweight. This material was utilized in a hybrid prosthetic mechanism for transfemoral amputees [18].

- Advanced or 'smart' materials

Magnetorheological fluids: These fluids are integrated into adaptive control systems to modulate mechanical properties in real time. Although this material has not yet been used in parallel robots, it is important to note that it can be utilized human prosthesis [116].

Shape-memory alloys (SMAs): Predominantly used in the design of actuators and responsive joints, these alloys enable a broad range of functional capabilities. Although this material has not yet been used in parallel robots, it is important to note that it can be utilized in robotic and biomedical applications [117].

Piezoelectric materials: These are incorporated in systems requiring ultra-precise micromovements and sensing applications. Although this material has not yet been used in parallel robots, it is important to note that it can be utilized in robotic and biomedical applications [118].

- Soft materials

Silicone: Predominantly employed in soft robotics, silicone's deformable yet resilient nature confers flexibility to the robot design. This material was used in a soft robotic neck that employed a parallel robot [3].

Elastomers: Capable of undergoing shape or stiffness alterations in response to electrical or thermal stimuli, elastomers are gaining traction in soft robotic applications. This material was utilized in a parallel lower limb based on pneumatic artificial muscles [22] and a soft parallel robot based on pneumatic artificial muscles for wrist rehabilitation [63].

*5.7. Critical Technological Challenges and Future Prospects in Rehabilitation, Assistance, and Humanoids*

There are considerable challenges that require keen attention for parallel robot technologies to transcend from research labs to commercial markets.

- Key technological challenges

Precision and reliability: Both rehabilitation and assistive robotics demand highly precise and reliable systems. The challenge extends to humanoid robots, which must interact safely and effectively in human-centric environments. The need for high precision and fail-safe operation in medical rehabilitation sets a challenging bar for

system performance. Addressing issues related to sensor accuracy and the reliability of actuators is of paramount importance.

User interface and experience: As these systems are ultimately designed for human use, developing intuitive, user-friendly interfaces that can cater to a diverse demographic is a significant hurdle. The end-users range from medical professionals to patients and the general public. Designing interfaces that are universally intuitive is an ongoing challenge.

Material constraints: Material choices should balance durability, cost, and biocompatibility, especially for wearable devices. Ensuring that all materials and operational modalities are biocompatible, sterile, and safe for long-term use requires rigorous testing and often leads to increased costs.

Scalability and affordability: Creating systems that are economically viable to manufacture at scale while maintaining high-quality performance presents an ongoing challenge.

- Impediments to commercialization

  Regulatory hurdles: Approvals for medical and assistive devices are lengthy and require substantial investment in clinical trials and validation studies.

  Cost factor: Advanced materials and smart systems drive up the manufacturing costs, making it difficult to position these products as affordable solutions.

- Future directions

  To realize the full potential of parallel robotic systems in the rehabilitation sector, future work needs to focus on solving the technological issues mentioned above. Solutions may lie in material innovations and machine learning algorithms for adaptive control. This would facilitate commercial transitions but also pave the way for more effective and accessible rehabilitation and assistance technologies.

## 6. Conclusions

- The data presented in Table 8 highlight the global interest and collaborative efforts in researching parallel robots for rehabilitation across various joint areas. China, the USA, and several other countries have actively contributed to this burgeoning field, each with a specific focus on particular joints. This comprehensive effort underscores the significance of parallel robot technology in furthering our understanding of human movement and enabling assistance across various joints.

- As the field of robotics continues to evolve, international collaboration can collectively drive advancements and shape the future of robotic technologies in rehabilitation, assistance, and humanoid applications.

- Technological development is a multifaceted process influenced by numerous factors. In the case of parallel robots, several challenges may have impeded further advancement. These could include technical limitations requiring complex and advanced solutions, as well as the inherent complexity of integrating sophisticated mechanical and electronic systems and control algorithms. Meeting these challenges necessitates ongoing research and development efforts.

- Parallel mechanisms offer multiple advantages over their serial counterparts, including greater rigidity, load capacity, precision, and speed. These characteristics make them well-suited for applications requiring high precision, such as rehabilitation and assistance in the neck, shoulder, wrist, hip, and ankle joints. Due to their design and configuration, parallel robots can distribute loads across their kinematic chains, enabling the manipulation of heavier objects and tasks requiring greater force.

- Parallel robots hold significant potential for rehabilitation, assistance, and humanoid applications, but many challenges and opportunities remain. These include the need for advanced sensors and control systems to improve accuracy and minimize movement errors.

- Looking to the future, continuous technological advancement and collaboration among the scientific community, industry, and healthcare professionals could make parallel robots a mainstay in the market. Their applications could extend to a wide range

of sectors, including but not limited to the automotive, aerospace, and construction industries, as well as military training, medicine, and surgery.

**Author Contributions:** V.E.A. contributed to the conceptualization, methodology, research, formal analysis, resources, and writing of the original draft. D.A.E. contributed to the conceptualization, review, editing, and supervision. All authors have read and agreed to the published version of the manuscript.

**Funding:** This research received no external funding.

**Conflicts of Interest:** The authors declare no conflict of interest.

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
