# Peer review of "A Review of Parallel Robots: Rehabilitation, Assistance, and Humanoid Applications for Neck, Shoulder, Wrist, Hip, and Ankle Joints"

_robotics, doi:10.3390/robotics12050131_

Round 1

Reviewer 1 Report

This review article presents research and development in robotics relevant to rehabilitation, assistive technologies, and humanoids using parallel mechanisms for human body joints with three degrees of freedom: neck, shoulder, wrist, hip, and ankle. Several flaws exist in this paper.

1.     Reviewer believe that the abstract should not simply list the relationship between the number of papers and the replaced joints but the classification of the replaced functions of the 3-DOF parallel mechanism. It is suggested to add a discussion of the connection between the alternative function and the configuration of the 3-DOF parallel mechanism in the abstract section.

2.     Workspace, function, and performance evaluation methods should be compared and evaluated in the literature cited.

3.     Reference 6 should be 2018,5,13. Reference 10 does not ×××. Reference 81 should be 2017,139 (2), 021608.

Reviewer 2 Report

This manuscript presents a review regarding the use of parallel robots in healthcare-related and humanoid applications. The manuscript is easy to follow and the research methodology is adequate. The review summarizes the reviewed works in proper tables and discuss the most relevant papers. The Discussion section is interesting and transmits a general feeling on the current state of the field and on how the field has evolved in the last decade. This reviewer finds it rather odd that the review only included articles from 2012-2022, excluding the current year. Nevertheless, I believe that this manuscript will be a positive addition to the literature.

Reviewer 3 Report

This review article introduces a state-of-the-art on the development in robotics related to rehabilitation, assitive technologies and humonoids friendly mechanisms for neck, shoulder, wrist, etc. This paper is fairly organized with an appropriate section, but the authors did not consider many other rehabilitation systems or devices published in the relevant journals:

·         Archives of Physical Medicine and Rehabilitation.

·         Disability and Rehabilitation.

·         Clinical Rehabilitation.

·         American Journal of Physical Medicine and Rehabilitation.

·         Journal of Head Trauma Rehabilitation.

·         Journal of Rehabilitation Medicine

·         Clinical Rehabilitation

·         Topics in Stroke Rehabilitation

·         JMIR Rehabilitation and Assistive Technologies

·         Journal of Sport Rehabilitation

·         Human Reproduction

·         Soft Robot

The authors shoud survey more articles from the above journals and add to the right position.

In addititon, this reviewer recommends the authors to arrange the review articles in a chronological manner to understand the development history of the robot assisted rehabilitation.

Recently, there are many rehabilitation systems or devices utilizing smart materials such as magnetorheological fluids, shape memory alloys and others. This a new technique for the development of advanced rehabilitation systesm and hence the authors should include this research field. 

Finally, the most significant issue (technology) of different rehabiltation system needs to be discussed in detail and the impediments to realize for commercial products also seriously addressed as future challanging works.

English should be improved a lot.

Reviewer 4 Report

The paper presents a review of parallel robots for Rehabilitation, Assistance, and Humanoid Applications, focusing on the joints used for neck, shoulder, wrist and ankle.

 point 1)

Section 4 presents a trivial discussion of what the robots are and a trivial comparison between serial, parallel and hybrid robots. The paper focuses on the usage of parallel robots in the fields of interest of the review, and this section does not introduce scientific contributions to the paper. 

The authors must remove the section 4.

 point 2)

Section 6.2 discusses the TRL of the applications found in the research articles investigated by the authors' review. The section also presents the definition of the TRL in lines 882-884 and in Table 10. 

The definition of TRL is well known and does not matter for the target of the article. 

The author must remove table 10 and any reference to it.

 point 3)

In lines 964-982, the paper discusses the numerical tools used by the articles investigated by the authors' review to simulate parallel robots. Here, the authors introduce comments and some comparing arguments on the numerical tools' capabilities for solving parallel robot kinematic and dynamic problems, referencing the authors' review papers. The presented discussion on the software tools is trivial regarding the software's capabilities and is based on references concerning the software's application rather than the software's characteristics.

The authors must rewrite this part of the article, introducing a more scientific approach and discussion.

 point 4)

The article's discussion and conclusion sections do not introduce a comparison of the authors' review papers based on their scientific contents. Moreover, the authors do not present the limitations of the existing study with future perspectives.

Instead, the authors propose a comparison of the papers based on statistical data regarding the authors' country or TRL of the robot's application.

The authors must introduce a comprehensive discussion of the review papers based on their scientific results, highlighting the scientific trends, for instance, based on the number of degrees of freedom or kinematic structure used for the joints.

point 5)

Regarding the paper's introduction and the discussion of the dynamic performances of parallel kinematic machines, the reviewer found more recent articles discussing the topic, so the reviewer recommends enriching the literature review in the paper's introduction by citing the additional references:

  • Modal kinematic analysis of a parallel kinematic robot with low-stiffness transmissions
  • Experimental set-up for the investigation of transmissions effects on the dynamic performances of a linear PKM

The paper does contain some awkward English phrasing and could benefit from a review with an English specialist. For instance, in line 44:

"However, a review of the current literature on the use of parallel robots in rehabilitation, assistance, and humanoids."

Round 2

Reviewer 3 Report

This paper has been well revised based on the reviewers' comments. Thus, it is now acceptable in its current form.

English needs to be polished a little.

Reviewer 4 Report

The paper can be accepted and published in the actual form.